# Scaling Curriculum Learning for Autonomous Driving

## Abstract

Batched simulators for autonomous driving have recently enabled the training of reinforcement learning agents on a massive scale, encompassing thousands of traffic scenarios and billions of interactions within a matter of days. Although such high-throughput feeds reinforcement learning algorithms faster than ever, their sample efficiency has not kept pace: As the standard training scheme, domain randomization uniformly samples scenarios and thus consumes a vast number of interactions on cases that contribute little to learning. Curriculum learning offers a remedy by adaptively prioritizing scenarios that matter most for policy improvement. We present CL4AD, the first integration of curriculum learning into batched autonomous driving simulators by framing scenario selection as an unsupervised environment design problem. We introduce utility functions that shape curricula based on success rates and the realism of the agent's behavior, in addition to existing regret-estimation functions. Large-scale experiments on GPUDRIVE demonstrate that curriculum learning can achieve 99% success rate a billion steps earlier than domain randomization, reducing wall clock time by 77%, and by 40% compared to traffic density-based heuristic curricula. An ablation study with a computational budget further shows that curriculum learning improves sample efficiency by 67% to reach the same success rate. To support future research, we release an implementation of CL4AD in GPUDRIVE.

## 1 Introduction

Batched simulators for autonomous driving (AD) have recently empowered sample-inefficient but effective reinforcement learning (RL) algorithms by enabling training for billions of interactions within a few days (Cusumano-Towner et al., 2025; Kazemkhani et al., 2025). These simulators achieve such scale by training RL agents on hundreds to thousands of scenarios in parallel through self-play (Silver et al., 2017), where a single policy controls all vehicles, taking millions of actions per second. For example, agents trained on GPUDRIVE (Kazemkhani et al., 2025) using the Waymo Open Motion Dataset (WOMD) (Ettinger et al., 2021) reliably generalize to unseen test cases in less than a day. GIGAFLOW (Cusumano-Towner et al., 2025), further scales self-play to 1.6 billion kilometers of simulated driving within 10 days, producing generalist driving policies that outperform benchmark-specific agents on CARLA (Dosovitskiy & Koltun, 2016), nuPlan (Caesar et al., 2021), and Waymax (Gulino et al., 2023) without any training on these benchmarks.

Despite advances in high simulation throughput, training RL agents in batched driving simulators remains sample-inefficient due to a standard training strategy: uniform scenario sampling, i.e., domain randomization (DR). This approach wastes a massive number of interactions on scenarios that are either too easy to provide a sufficient learning signal or too difficult for the current policy to make progress on. Curriculum learning (CL) offers a remedy by adaptively prioritizing scenarios that contribute the most to policy improvement (Narvekar et al., 2020). In particular, curriculum learning has successfully fulfilled that promise in multiple large-scale RL domains. For example, Bauer et al. (2023) demonstrate that scaling meta-RL with automated curricula yields agents capable of human-timescale adaptation across thousands of procedurally generated environments. Zhang et al. (2024) introduce curricula for open-ended environments, where there are infinitely many possible tasks, showing that curriculum learning enables faster and broader skill acquisition.

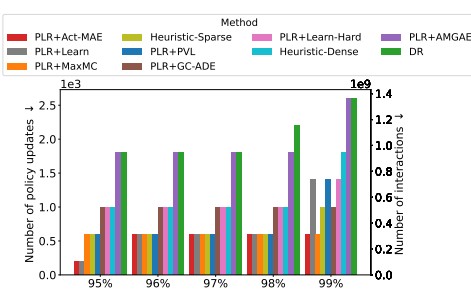

Figure 1: CL4AD integrates UED methods into a batched AD simulator to adaptively prioritize traffic scenarios based on three types of utility functions: regret, success, and realism.

Inspired by the success of CL in large-scale RL, we introduce CL4AD (see Fig. 1), the first integration of automated curricula into batched AD simulators. We frame scenario selection as an unsupervised environment design problem (UED), and equip prioritized level replay (PLR) (Jiang et al., 2021b) with utility functions that adaptively shape training. Therefore, curricula prioritize scenarios at the frontier of the agent's capabilities, rather than relying on uniform sampling. Across large-scale experiments, we show that CL accelerates RL training by hundreds of millions of steps compared to DR and heuristic curricula. Our key contributions are below:

Figure 2: PLR achieve 99% success rate in one billion steps before DR. For acronyms of utility functions, see Sections 3.3 and 4.

- We present **CL4AD**, the first integration of curriculum learning methods from unsupervised environment design to the scale of GPU-accelerated, self-play simulators for AD, and provide an implementation in an open-source batched AD simulator, GPUDRIVE.
- We propose **novel utility functions** based on the success and realism of agent behavior.
- We conduct a **large-scale empirical study** 1) showing that CL accelerates RL training by up to a billion interactions, improving the sample-efficiency to reach 99% success rate by 77% compared to DR and by 40% compared to traffic density-based heuristic curricula (see Fig. 2); 2) illustrating the effects of utility functions on learning; 3) investigating the effectiveness of CL under limited resources; and 4) analyzing the correlation between utility functions and performance metrics.

## 2 RELATED WORKS

**Autonomous driving simulators** have enabled RL to train self-driving agents in multiple ways: Simulators such as Waymax (Gulino et al., 2023) and Nocturne (Vinitsky et al., 2022) use traffic scenarios from open-source driving datasets such as WOMD (Ettinger et al., 2021), whereas CARLA (Dosovitskiy et al., 2017) is not data-driven, and Metadrive enables procedural scenario generation as well as integration of real driving data. Attempts to scale RL for AD have resulted in batched simulators such as Waymax, GPUDRIVE (Kazemkhani et al., 2025), and GIGAFLOW (Cusumano-Towner et al., 2025), which significantly increased the data throughput to feed RL algorithms. These simulators so far have utilized random scenario generation/sampling to train AD agents.

**Curriculum learning for RL** accelerates learning optimal policies by sequencing different configurations of the environment with respect to the capabilities of the trained agent (Narvekar et al., 2020). Automated curriculum generation studies goal-conditioned domains (Baranes & Oudeyer, 2010; Florensa et al., 2018; Tzannetos et al., 2023), contextual settings (Klink et al., 2022; Koprulu et al., 2023; Sayar et al., 2024), and more popularly UED (Dennis et al., 2020). UED models environments with free parameters, calling an instance a *level*. UED methods generate levels via trained teacher agents, e.g., PAIRED (Dennis et al., 2020) and RE-PAIRED (Jiang et al., 2021a), or by randomly sampling free parameters, e.g., in PLR and ACCEL (Parker-Holder et al., 2022). PLR, as one of the earlier UED approaches, has shown evidence of scalability in settings such as meta RL (Bauer et al., 2023)(Jackson et al., 2023), and open-ended environments (Zhang et al., 2024).

**Curriculum learning for AD** aims to speed up training self-driving policies via RL, e.g., ScenarioNet (Li et al., 2023), which unifies heterogenous data for traffic simulation, showcases benefits of heuristic-based curricula. Similarly, Anzalone et al. (2021; 2022) propose a multi-stage curriculum learning method for CARLA, incrementally making the number of agents, their initial positions, or weather conditions incrementally more difficult. In contrast to manual curricula, Qiao et al. (2018) develops an automated method for urban intersections. Recently, Brunnbauer et al. (2024) and Abouelazm et al. (2025) have demonstrated that UED methods, RE-PAIRED and ACCEL, respectively, accelerate training AD agents in CARLA. However, there has not been any investigation into whether CL can scale with the high throughput and scenario diversity enabled by batched simulators such as GPUDRIVE, which trains self-driving RL agents in scenarios from real driving datasets.

## 3 BACKGROUND

We model a traffic scenario as a *partially observable stochastic game* (POSG), similar to Brunnbauer et al. (2024), to accommodate the multi-agent nature of driving. Upon defining this model, we explain how batched AD simulators GIGAFLOW and GPUDRIVE train self-driving agents via a multi-agent RL scheme called *self-play* in traffic scenarios. Then, we frame curriculum learning for autonomous driving as *unsupervised environment design*, and describe how to measure the utility of traffic scenarios to improve sample-efficiency. Lastly, we illustrate a popular UED method called *prioritized level replay* (PLR), the backbone of the curriculum learning algorithm in our work.

### 3.1 TRAFFIC SCENARIOS AS PARTIALLY OBSERVABLE STOCHASTIC GAMES

**Definition 3.1.** *A POSG is a tuple* $\mathcal{G} = \langle \mathcal{N}, \mathcal{S}, \mathcal{A}, \mathcal{O}, T, Z, R, I, \gamma \rangle$*, where* $\mathcal{N} = [N]$ *is the set of agents with* $N \in \mathbb{Z}^+$*,* $\mathcal{S}$ *is the state space,* $\mathcal{A} = \mathcal{A}_1 \times \mathcal{A}_2 \times \cdots \times \mathcal{A}_N$ *and* $\mathcal{O} = \mathcal{O}_1 \times \mathcal{O}_2 \times \cdots \times \mathcal{O}_N$ *are the joint action and observation spaces.* $T : \mathcal{S} \times \mathcal{A} \to \Delta(\mathcal{S})$ *represents the stochastic dynamics of a POSG, i.e, the probability of transitioning from state* $\mathbf{s} \in \mathcal{S}$ *to state* $\mathbf{s}' \in \mathcal{S}$ *given joint action* $\mathbf{a} \in \mathcal{A}$*.* $Z : \mathcal{S} \times \mathcal{A} \to \Delta(Z)$ *determines the probability of observing* $\mathbf{o} = (\mathbf{o}_1, \mathbf{o}_2, \cdots, \mathbf{o}_N) \in \mathcal{O}$ *in state* $\mathbf{s}$ *taking joint action* $\mathbf{a}$*. The reward function* $R : \mathcal{S} \times \mathcal{A} \to \mathbb{R}^N$ *determines rewards, namely,* $R(\mathbf{s}, \mathbf{a}) = (R_1(\mathbf{s}, \mathbf{a}), R_2(\mathbf{s}, \mathbf{a}), \cdots, R_N(\mathbf{s}, \mathbf{a}))$ *where* $R_i(\mathbf{s}, \mathbf{a}) \in \mathbb{R}$ *is the reward for agent* $i \in \mathcal{N}$*.* $I \in \Delta(\mathcal{S})$ *represents the initial state distribution. Finally,* $\gamma \in [0, 1]$ *is the discount factor.*

A policy $\pi_i : \mathcal{O}_i \to \Delta(\mathcal{A}_i)$ describes the behavior of agent $i$ in POSG $\mathcal{G}$. The value function for $\pi_i$ is the expected cumulative discounted rewards over a horizon of H steps, i.e., $V(\pi_i) = \mathbb{E}_{T,Z} \left[ \sum_{t=0}^{H-1} \gamma^t R_i(\mathbf{s}_t, \mathbf{a}_t) | \mathbf{s}_o \sim I, \mathbf{a}_t = (\mathbf{a}_{j,t})_{j \in \mathcal{N}} \right]$ where $\mathbf{a}_{j,t} \sim \pi_j(\mathbf{o}_{j,t})$. Agent $i$ aims to find an optimal policy $\pi_i^*$, which maximizes its value $V(\pi_i)$ in POSG $\mathcal{G}$.

In a traffic scenario modeled as $\mathcal{G}$, consider $\pi_i$ as a policy that controls vehicle $i$. The road layout, traffic rules, and collision dynamics in a scenario specify the dynamics $T$. Initial state $\mathbf{s}_0 \sim I$ consists of the initial positions of all vehicles, pedestrians, cyclists, etc. Observation $\mathbf{o}_{i,t}$ of vehicle $i$ at time $t \in [H]$ is what the controller perceives about the surroundings based on its sensors as well as specific attributes, e.g., the type of vehicle, its velocity, acceleration, etc. The reward $r_i = R_i(\mathbf{s}_t, \mathbf{a}_t)$ can incentivize the policy to reach a goal location, stay within lanes, and avoid collisions.

To model multiple traffic scenarios, following Brunnbauer et al. (2024), we formalize a set of traffic scenarios as an *underspecified* POSG (UPOSG), which captures a set of traffic scenarios.

**Definition 3.2.** *An underspecified POSG* $\mathcal{G}^\Theta = \langle \Theta, \mathcal{N}^\Theta, \mathcal{S}, \mathcal{A}^\Theta, \mathcal{O}^\Theta, T^\Theta, Z^\Theta, R^\Theta, I^\Theta, \gamma \rangle$ *models a set of POSGs through parameters* $\theta \in \Theta$ *that determine the set of agents* $\mathcal{N}^\Theta$*, and all attributes of a POSG* $\theta \in \Theta$ *depending on its agents, such as the dynamics* $T^\Theta : \mathcal{S} \times \mathcal{A}^\Theta \times \Theta \to \Delta(\mathcal{S})$*.*

Consider scenarios $\Theta = \{\theta_m\}_{m \in [M]}$ in WOMD (Ettinger et al., 2021), where M $\approx 100,000$. A scenario $\theta_m$ may correspond to an urban intersection, a parking lot, or a highway, with varying speed limits, number of vehicles, etc. In practice, $\theta_m$ is merely an identification number, i.e., $\theta_m \in [M]$, hence it does not reveal such properties of the scenario, which makes it underspecified.

### 3.2 SELF-PLAY RL IN BATCHED AUTONOMOUS DRIVING SIMULATORS

Self-play RL is an RL scheme for multi-agent settings where each agent samples their actions from a shared, decentralized policy. More formally, this scheme samples the action $\mathbf{a}_{i,t} \sim \pi_\phi(\mathbf{o}_{i,t})$ of

agent $i \in \mathcal{N}^{\Theta}$ via a policy $\pi_{\phi}$ parameterized by $\phi$, e.g., a neural network with learnable parameters $\phi$, given the observation $\mathbf{o}_{i,t}$ of said agent at time $t$. Batched AD simulators GIGAFLOW and GPUDRIVE use self-play RL as the strategy to train a single policy that controls all vehicles in a scenario in parallel. Their batched structure empowers parallelization further by concurrently simulating hundreds to thousands of traffic scenarios to accelerate experience collection. Both works employ an on-policy RL algorithm, proximal policy optimization (PPO) (Schulman et al., 2017), where policy updates occur once the simultaneous data collection fills an experience buffer. As a result, batched simulation accelerates experience collection via parallelized scenarios, while self-play RL saves compute time and memory by training a single policy. We implement CL4AD on GPUDRIVE, which samples hundreds of traffic scenarios every couple of million interactions, with initial positions and goals from logged traffic data in WOMD. The default scenario sampling is uniformly random, i.e., via domain randomization, where every traffic scenario has equal likelihood.

## 3.3 UNSUPERVISED ENVIRONMENT DESIGN

UED (Dennis et al., 2020) aims to generate a sequence of *levels*, i.e., traffic scenarios $\theta \in \Theta$ in the case of AD, to accelerate learning a policy that generalizes across all levels [1]. One solution to UED is a level generator $\Lambda : \Pi \to \Delta(\Theta)$ that produces a distribution over the set of all levels $\Theta$ given a policy $\pi \in \Pi$. A level generator $\Lambda$ maximizes some utility function $U(\pi, \theta)$ that measures the contribution of a level $\theta$ to sample efficiently improve $\pi$. Without loss of generality, in this section, we assume that there is only one agent in a level $\theta$, i.e., $\mathcal{N}^{\Theta} = [1]$, to ease the use of notation.

Domain randomization, i.e., uniformly sampling levels throughout the training, is the default way of training an RL agent where the utility is constant for each level, namely, $U(\pi, \theta) = C$, $\theta \in \Theta$ and $C \in \mathbb{R}$. UED methods primarily differ in their utility functions of choice, as it is not possible to accurately calculate the contribution of all levels to policy improvement. There are two common categories of utility functions: regret and success-based. Regret, i.e., the difference between the expected discounted return of the current policy and the optimal one, is a convenient objective as a level generator that maximizes regret will prioritize the easiest levels that the agent cannot currently solve (Dennis et al., 2020). More formally, a regret-based utility is $U^{\text{Regret}}(\pi, \theta) = V^{\theta}(\pi_{\theta}^{*}) - V^{\theta}(\pi)$, where $\pi_{\theta}^{*}$ is an optimal policy in level $\theta$, i.e., a policy collecting the maximum expected discounted return $V^{\theta}(\pi_{\theta}^{*})$. However, as the optimal expected discounted return or the optimal policy for each level is rarely available, UED methods estimate regret in various ways. Jiang et al. (2021b) propose learning potential, i.e., *average magnitude of the generalized advantage estimate* (AMGAE) (Schulman et al., 2015) as a utility function that estimates regret over a single episode,

$$U^{\text{AMGAE}}(\pi, \theta) = \frac{1}{H} \sum_{t=0}^{H-1} \left| \sum_{k=t}^{H-1} (\gamma\lambda)^{k-1} \delta_k \right|, \tag{1}$$

where $\delta_k = r_k + \gamma V^{\theta,\pi}(\mathbf{o}_{k+1}) - V^{\theta,\pi}(\mathbf{o}_k)$ is the temporal difference error at timestep $k$, $V^{\theta,\pi}(\mathbf{o}_k) = \mathbb{E}_{T^{\Theta}, Z^{\Theta}} \left[ \sum_{t=k}^{H-k-1} \gamma^{t-k} R^{\Theta}(\mathbf{s}_t, \mathbf{a}_t, \theta) | \mathbf{a}_t \sim \pi(\mathbf{o}_t) \right]$ is the expected discounted return of $\pi$ in $\mathbf{s}_k$ on level $\theta$, and $\lambda$ is the discount factor for GAE. Alternatively, Jiang et al. (2021a) and Parker-Holder et al. (2022) employ *positive value loss* (PVL), i.e.,

$$U^{\text{PVL}}(\pi, \theta) = \frac{1}{H} \sum_{t=0}^{H-1} \max \left\{ \sum_{k=t}^{H-1} (\gamma\lambda)^{k-1} \delta_k, 0 \right\}. \tag{2}$$

As PVL uses the bootstrapped value target to compute the temporal difference error, Jiang et al. (2021a) also propose *maximum Monte Carlo* (MaxMC), which instead utilizes the highest return obtained by $\pi$ on level $\theta$ to mitigate potential bias issues,

$$U^{\text{MaxMC}}(\pi, \theta) = \frac{1}{H} \sum_{t=0}^{H-1} \left( R_{\max}^{\theta} - V^{\theta,\pi}(\mathbf{o}_t) \right), \tag{3}$$

where $R_{\max}^{\theta}$ is the maximum discounted return achieved in level $\theta$ so far during training.

Success-based utility functions address settings where a level $\theta$ is considered solved when a policy $\pi$ reaches a goal state $\mathbf{s} \in \mathcal{S}_{\text{Goal}}^{\theta} \subset \mathcal{S}$. Such utility functions use the success rate $p^{\theta,\pi}$, i.e., the fraction

---

[1] As *level* is the common term in the UED literature to describe $\theta$, we use it interchangeably with *scenario*.

of times policy $\pi$ solves a level $\theta$, $p^{\theta,\pi} = \mathbb{P}\left(\exists t \in [\text{H}] : \mathbf{s}_t \in \mathcal{S}_{\text{Goal}}^{\theta} | \pi, \theta\right)$. Inspired by Tzannetos et al. (2023), Rutherford et al. (2024) propose *Sampling for Learnability* (SFL) along with *learnability*

$$U^{\text{Learn}}(\pi, \theta) = p^{\theta,\pi} \cdot (1 - p^{\theta,\pi}), \tag{4}$$

a utility function that can be interpreted as the variance of a Bernoulli distribution with parameter $p^{\theta,\pi}$, namely, how inconsistent policy $\pi$ is at solving $\theta$. Rutherford et al. (2024)'s analysis reveals that in sparse reward settings, where only non-zero rewards occur when a policy reaches the goal, regret-based utility functions have low correlation with the success rate. They argue that regret-based utility functions become noisy in such settings, causing inaccurate identification of the learning frontier. Therefore, learnability is useful, especially for autonomous driving, where reward functions commonly reward and punish sparse events such as goal completion and collisions, respectively.

### 3.4 PRIORITIZED LEVEL REPLAY

*Prioritized Level Replay* (Jiang et al., 2021b) is one of the first UED methods that lays the foundation for approaches such as Robust PLR (Jiang et al., 2021a), REPAIRED, ACCEL, and SFL. PLR consists of two steps: uniformly sampling levels from a set of training levels $\Theta^{\text{train}}$, and replaying levels from a rolling buffer $\mathcal{B}$. At the beginning of the training, PLR evaluates the agent on randomly sampled levels, and scores these levels using regret-estimating utility function $U^{\text{AMGAE}}$ Eq. (1). Then, PLR adds levels with the highest scores to its buffer. Subsequently, PLR makes a random decision with probability $d$ to sample unseen levels in $\Theta^{\text{train}}$ or seen levels from the buffer via a distribution based on their scores and staleness, namely,

$$\mathbb{P}_{\text{replay}}(\theta_i | \mathcal{B}, U^{\text{AMGAE}}, l) = (1 - \rho) \cdot \mathbb{P}_{\text{utility}}(\theta_i | \mathcal{B}, U^{\text{AMGAE}}) + \rho \cdot \mathbb{P}_{\text{staleness}}(\theta_i | \mathcal{B}, l), \tag{5}$$

where $\mathbb{P}_{\text{utility}}(\theta_i | \mathcal{B}, U^{\text{AMGAE}})$ is based on the ranking of seen levels with respect to their scores, i.e.,

$$\mathbb{P}_{\text{utility}}(\theta_i | \mathcal{B}, U^{\text{AMGAE}}) = \frac{\text{rank}(\theta_i | \mathcal{B})^{-1/\beta}}{\sum_{j \in \mathcal{B}^{\text{scenario}}} \text{rank}(\theta_j | \mathcal{B})^{-1/\beta}}, \tag{6}$$

with a temperature parameter $\beta$ tuning the impact of ranking. The staleness distribution assigns a higher likelihood for levels that has not been sampled for a higher number of episodes, namely,

$$\mathbb{P}_{\text{staleness}}(\theta_i | \mathcal{B}, l) = \frac{l - l_{\theta_i}}{\sum_{j \in \mathcal{B}^{\text{scenario}}} l - l_{\theta_j}}, \tag{7}$$

where $l$ is the total number of levels sampled so far, and $l_{\theta_j}$ is the episode count at which level $\theta_j$ was last sampled. This distribution aims to prevent the scores of seen levels from becoming off-policy, as they may remain in the buffer for a while without being sampled during training. Note that Robust PLR and SFL have similar buffer and sampling mechanisms with PLR, except that Robust PLR does not update the policy using rollouts from unseen levels and SFL has a filtering mechanism that requires additional rollouts to assess whether a level has high learnability.

## 4 CURRICULUM LEARNING FOR AUTONOMOUS DRIVING AT SCALE

*Curriculum learning for autonomous driving*, CL4AD, integrates variants of an existing UED method, PLR, into a batched AD simulator by scaling them up in terms of four aspects: **(1)** Concurrent simulation of hundreds of traffic scenarios, **(2)** Tracking tens of agents in a single scenario, **(3)** Training in tens of thousands of scenarios, and **(4)** Training for billions of steps. To adapt a UED method to a batched simulator, CL4AD tracks the behavior of all self-play agents in all concurrent scenarios. For example, in GPUDRIVE, where we implement CL4AD, simulated scenarios come from real-world datasets, and each scenario has a specific horizon H due to the nature of the logged data. CL4AD treats each scenario as a separate $\theta_i \in \Theta$ to enable measurement and tracking of their utility. Between scenario sampling steps, CL4AD monitors each simulated scenario and computes its utility once an episode terminates, which occurs when all agents reach their goals, collide, or the time exceeds the horizon. In essence, the utility of a traffic scenario corresponds to the expected performance of a self-play policy that controls all agents in the scenario, thereby capturing the expected collective behavior. To address the multi-agent aspect, we make a change in the definition of utility functions in Section 3.3, e.g., we formally define $U^{\text{MaxMC}}$ as

$$U^{\text{MaxMC}}(\pi, \theta) = \mathbb{E}_{\pi,\theta}\left[\frac{1}{\text{H}} \sum_{t=0}^{\text{H}-1} \frac{1}{\text{N}_\theta} \sum_{n=1}^{\text{N}_\theta} \left(\mathbf{R}_{\max}^{\theta,n} - V^{\theta,\pi}(\mathbf{o}_{n,t})\right)\right], \tag{8}$$

---

**Algorithm 1** **C**urriculum **L**earning for **A**utonomous **D**riving (CL4AD)

---

**Input**: Set of training scenarios $\Theta^{\text{train}}$

**Parameters**: Replay rate $d$, Staleness coefficient $\rho$, temperature $\beta$, utility function $U$, max buffer size $\text{B}^{\text{max}}$, total number of iterations $\text{T}^{\text{train}}$, scenario sampling interval $\text{T}^{\text{sce}}$, policy update interval $\text{T}^{\text{pol}}$, number of worlds W

**Output**: Final policy $\pi_\phi$

1: $\mathcal{B} \leftarrow (), \mathcal{D} \leftarrow () \ t \leftarrow 0, l \leftarrow 0, \pi_\phi \leftarrow \pi_{\phi_0}$   ▷ *Reset scenario/experience buffers, iterators, and policy*
2: **while** $t < \text{T}^{\text{train}}$ **do**
3:   **if** $0 \equiv t \mod \text{T}^{\text{sce}}$ **then**
4:     $l \leftarrow l + 1$                             ▷ *Increment sampling iteration*
5:     $(\theta_w)_{w=1}^{\text{W}}, \mathcal{B} \leftarrow \text{SAMPLEFROMCURRICULUM}(\mathcal{B}, \Theta^{\text{train}}, l)$   ▷ *Sample scenarios for worlds*
6:     $\mathcal{D}_t = \{\{\mathbf{o}_{n,w}, \mathbf{a}_{n,w}, \mathbf{o}'_{n,w}, r_{n,w}, e_{n,w}\}_{n \in [\text{N}_{\theta_w}]}\}_{w \in [\text{W}]}$   ▷ *Record experiences over a single step*
7:     $\mathcal{B} \leftarrow \text{UPDATECURRICULUM}(\mathcal{D}_t, U, \mathcal{B})$   ▷ *Update curriculum with the scores of terminated scenarios*
8:     $\mathcal{D} \leftarrow \mathcal{D} \cup \mathcal{D}_t$                     ▷ *Update experience buffer with new interactions*
9:   **if** $0 \equiv t \mod \text{T}^{\text{pol}}$ **then**
10:     $\pi, \mathcal{D} \leftarrow \Phi(\mathcal{D})$     ▷ *Update self-play policy via RL algorithm $\Phi$, and reset the experience buffer $\mathcal{D}$*
11:   $t \leftarrow t + |\mathcal{D}_t|$                       ▷ *Update training iteration*

---

where $\text{R}_{\max}^{\theta,n}$ is the maximum return that agent $n \in [\text{N}_\theta]$ collected in scenario $\theta$ so far. In contrast to Eq. (3), Eq. (8) accounts for the expected behavior of $\pi$, as batched simulators enable the collection of multiple episodes in a scenario before sampling new scenarios. As an approximation, CL4AD computes the average of K-many episodes it observes between sampling steps. In addition to regret-based $U^{\text{AMGAE}}$, $U^{\text{PVL}}$, $U^{\text{MaxMC}}$, and success-based $U^{\text{Learn}}$, CL4AD introduces three novel utility functions: *learnability-hard* $U^{\text{Learn-hard}}$, *goal-conditioned average distance error* (GC-ADE) $U^{\text{GC-ADE}}$, and *action mean absolute error* (Act-MAE) $U^{\text{Act-MAE}}$, which we define as

$$U^{\text{Learn-hard}}(\pi, \theta) = \frac{1}{\text{N}_\theta} \sum_{n=1}^{\text{N}_\theta} p_{\text{hard}}^{\theta,\pi,n} \cdot (1 - p_{\text{hard}}^{\theta,\pi,n}), \tag{9}$$

$$U^{\text{GC-ADE}}(\pi, \theta) = \mathbb{E}_{\pi,\theta} \left[ \frac{1}{\text{N}_\theta} \sum_{n=1}^{\text{N}_\theta} \frac{1}{\text{H}} \sqrt{\sum_{t=0}^{\text{H}-1} \|\mathbf{x}_{n,t} - \mathbf{x}_{n,t}^{\text{logged}}\|_2^2} \right], \tag{10}$$

$$U^{\text{Act-MAE}}(\pi, \theta) = \mathbb{E}_{\pi,\theta} \left[ \frac{1}{\text{N}_\theta} \sum_{n=1}^{\text{N}_\theta} \frac{1}{\text{H}} \sum_{t=0}^{\text{H}-1} \|\mathbf{a}_{n,t} - \mathbf{a}_{n,t}^{\text{logged}}\|_1 \right]. \tag{11}$$

$U^{\text{Learn-hard}}$ is a success-based utility function that, in contrast to $U^{\text{Learn}}$, utilizes the rate of agent $n$ reaching its goal without colliding or going off-road in scenario $\theta$ via self-play policy $\pi$. Such difficult-to-satisfy success rates appear in AD works, as they capture both robustness and safety (Cusumano-Towner et al., 2025). $U^{\text{GC-ADE}}$ and $U^{\text{Act-MAE}}$ are realism-based utility functions that compute the distance between the positions and actions of RL agents and the logged trajectories, respectively. Since the fundamental objective in training AD policies is to deploy them in the real world, their realism becomes crucial for harmonious behavior. Realism-based metrics often serve as a way to evaluate behavior plausibility (Caesar et al., 2021; Gulino et al., 2023; Cornelisse & Vinitsky, 2024). In contrast, CL4AD uses them to determine which scenarios to prioritize.

Algorithm 1 is a pseudocode illustrating the integration of PLR into a batched simulator via CL4AD. At the beginning of the training, we initialize the parameters $\phi$ of the self-play policy $\pi_\phi$, and reset scenario and experience buffers $\mathcal{B}$ and $\mathcal{D}$, as well as the training and scenario sampling iterations, $t$ and $l$, respectively (Line 1). Until training iteration reaches $\text{T}^{\text{train}}$, CL4AD first checks if it is time to sample new scenarios via PLR based on its replay buffer $\mathcal{B}$ (Line 3-5). If so, CL4AD samples new scenarios, and sets them to concurrently simulated worlds. Note that PLR only keeps $\text{B}^{\text{max}}$ highest ranking scenarios in the buffer for sampling. Then, the self-play policy $\pi_\phi$ takes a step in all scenarios, and $\mathcal{D}_t$ records them (Line 6). CL4AD updates the curriculum buffer using the utility of terminated scenarios (Line 7). Note that each utility function requires different signals. For example, realism-based functions compare agents' observations/actions against logged data. Success-based ones check success and collision/off-road flag. Regret-based functions require rewards and values. Finally, an RL algorithm updates the policy using the experience buffer $\mathcal{D}$ (Line 8-10) every $\text{T}^{\text{pol}}$ steps. We refer the reader to Appendix D for more details on sampling from and updating curricula.

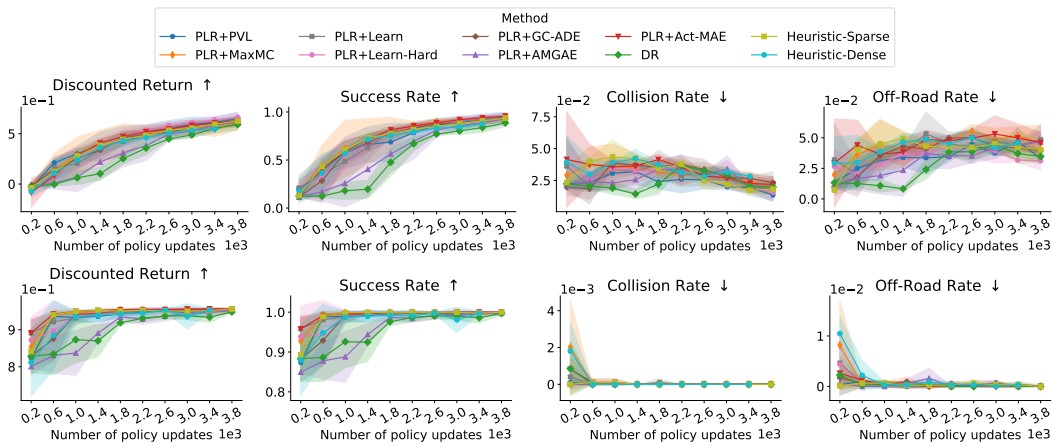

Figure 3: Case 1: Performance progression during training with 1000 scenarios: We evaluate in **(top)** training partition, and **(bottom)** 150 test scenarios. Bold markers indicate the mean, whereas the shaded area covers one standard deviation around it across three independent training runs.

## 5 EXPERIMENTAL RESULTS

We implement CL4AD in GPUDRIVE (Kazemkhani et al., 2025) and conduct experiments using traffic scenarios from WOMD (Ettinger et al., 2021) to investigate the following questions:

5.1) Can curriculum learning accelerate learning performant AD policies?
5.2) How does curriculum learning guide scenario selection?
5.3) Is curriculum learning effective under limited compute resources?
5.4) Can curriculum learning scale up with the number of scenarios?
5.5) Do utility functions correlate with each other and performance metrics?

For quantitative questions, we consider 1) return, 2) success, 3) collision, 4) off-road rates, and 5) goal-conditioned average displacement error (GC-ADE) (Cornelisse & Vinitsky, 2024) to assess performance, safety, and realism of trained policies. For qualitative questions, we visualize replay distributions, prioritized scenarios, and the progression of expected utility in training scenarios.

We train RL agents in GPUDRIVE using self-play PPO, following Kazemkhani et al. (2025); Cornelisse et al. (2025). The observation of an agent is its bird-eye-view (BEV) within a fixed radius, while its action consists of speed and steering inputs. Agents receive rewards for goal completion, and penalties for collisions and going off-road. Note that an episode does not terminate if a crash or off-road event occurs. We report results from CL4AD trained with PufferLib (Suarez, 2024). We compare DR, the default sampling approach, against two heuristic-based curriculum methods and **7 UED methods**, i.e., combinations of PLR with utility functions in Sections 3.3 and 4: Regret-based $U^{\text{AMGAE}}$, $U^{\text{PVL}}$ and $U^{\text{MaxMC}}$; success-based $U^{\text{Learn}}$ and $U^{\text{Learn-hard}}$; realism-based $U^{\text{GC-ADE}}$ and $U^{\text{Act-MAE}}$. Note that, by combining PLR with $U^{\text{PVL}}$, $U^{\text{MaxMC}}$ and $U^{\text{Learn}}$, we evaluate efficient versions of Robust PLR and SFL. Heuristic-Dense and Heuristic-Sparse prioritize scenarios with high and low vehicle counts, respectively. We refer the reader to Appendix E for more details.

### 5.1 CAN CURRICULUM LEARNING ACCELERATE LEARNING PERFORMANT AD POLICIES?

To evaluate curriculum learning in GPUDRIVE, we first train RL agents using a mini version of WOMD with 1,000 traffic scenarios and evaluate on the test partition with unseen 150 scenarios. Fig. 3 shows the progression of trained policies when evaluated on the training **(top)** and test **(bottom)** partitions. PLR, with all utility functions except $U^{\text{AMGAE}}$, achieves the highest returns and success rates in training scenarios. Fig. 2 further evidences that, in test scenarios, PLR achieves 99% success rate a billion steps earlier than DR, reducing wall clock time by 77%. Compared with Heuristic-Sparse and Heuristic-Dense, PLR accelerates training to achieve the same

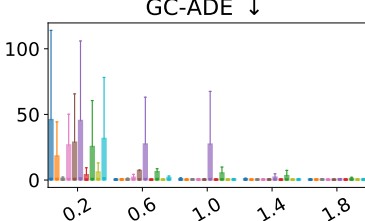

Figure 4: Case 1: Realism progression in test partition.

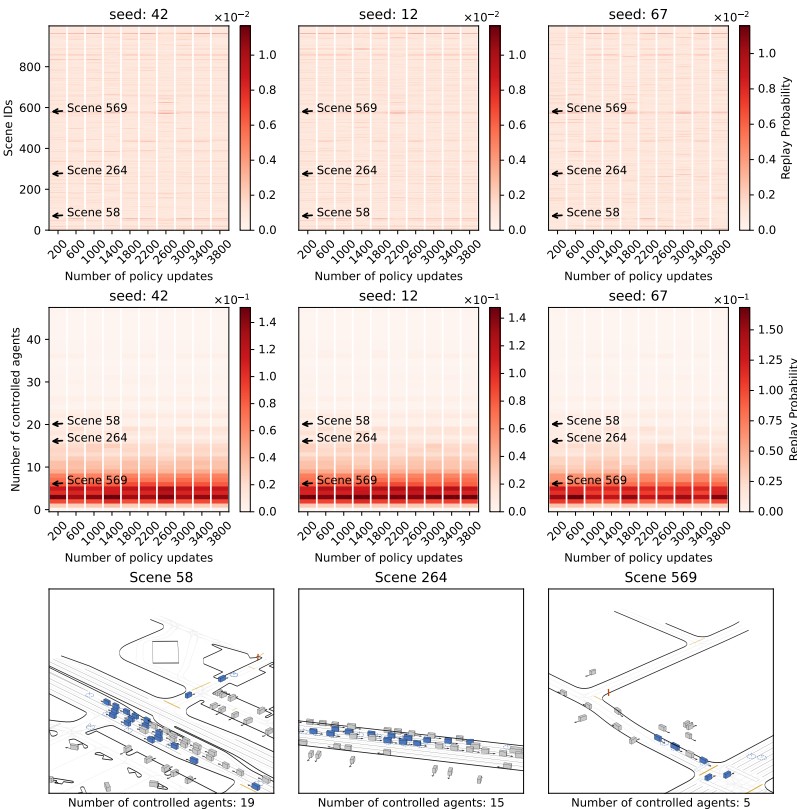

Figure 5: Case 1: $\mathbb{P}_{\text{replay}}$ progression of PLR combined with $U^{\text{MaxMC}}$ in mini WOMD: We illustrate **(top)** the evolution of $\mathbb{P}_{\text{replay}}$, where darker line segments indicate scenarios with higher replay likelihood, **(middle)** a version of replay distribution under categorization with respect to the number of controlled agents in scenarios, and **(bottom)** we exemplify three scenarios that appear frequently.

success rate by 40% and 66%, respectively. Note that PLR with $U^{\text{AMGAE}}$ outperforms DR with a small margin in terms of return. PLR also yields realistic policies faster than DR (see Fig. 4), showcasing that curriculum learning is not only sample-efficient but also obtains plausible behavior.

## 5.2 How does curriculum learning guide scenario selection?

Fig. 5 shows how replay distributions $\mathbb{P}_{\text{replay}}$ (Eq. (5)) of PLR combined with $U^{\text{MaxMC}}$ evolve across independent training runs. We observe that certain scenarios are consistently assigned a high likelihood (dark red) in all runs across multiple stages of training, such as those illustrated in the bottom row. Note that scenarios with ID 58 and 264 involve at least 15 controlled agents (blue), and highly congested cases are rare (see the middle row). Fig. 6 illustrates the progression of $U^{\text{Learn}}$ in training scenarios. Approaches that converge early (see Fig. 3), obtain high learnability early on, showing improved learning speed, and achieve the lowest learnability the fastest in the end.

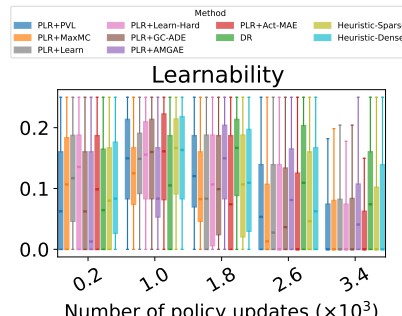

Figure 6: Case 1: Learnability.

## 5.3 Is curriculum learning effective under limited compute resources?

To investigate curriculum learning under computational constraints, we ran an ablation study using a GPU with significantly smaller memory, which only allows one-eighth of the number of worlds $W$ and one-fourth the size of the experience buffer $\mathcal{D}$, in contrast to the GPU we used in other cases

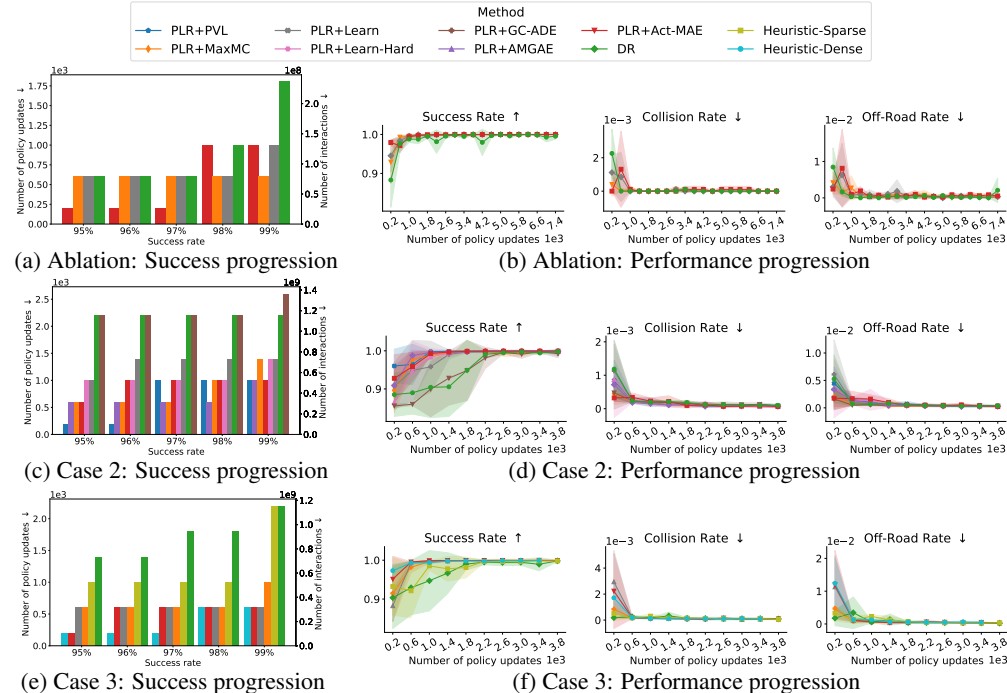

Figure 7: **(Top)** Ablation study on the effectiveness of CL under compute constraints across three independent runs. **(Middle-bottom)** Performance progression during training in **(case 2)** 10,000, and **(case 3)** 80,000, scenarios from WOMD, both evaluated in 10,000 unseen test scenarios.

(see Appendix F for more details). Although a smaller buffer results in a higher frequency of policy updates, this setup causes training to take about four times longer in wall-clock time while limiting the diversity of scenarios used for updates. Figs. 7a and 7b show that, although DR needs fewer interactions than the regular set-up, PLR is faster at reaching 99% success rate by 67% than DR.

## 5.4 CAN CURRICULUM LEARNING SCALE UP WITH THE NUMBER OF SCENARIOS?

To evaluate the scalability of curriculum learning for AD in terms of training dataset size, we train self-play agents in (case 2) 10,000 and (case 3) 80,000 scenarios from WOMD. Figs. 7c and 7d demonstrate that, PLR reduces the number of interactions needed to reach 99% success rate by over 55%, when combined with $U^{\text{MaxMC}}$ and $U^{\text{Act-MAE}}$ in case 2. Similarly, Figs. 7e and 7f show that PLR improves sample-efficiency by 72% when combined with $U^{\text{Learn}}$ in case 3. Here, Heuristic-Dense can match PLR, whereas Heuristic-Sparse does not have any advantages over DR.

## 5.5 DO UTILITY FUNCTIONS CORRELATE WITH EACH OTHER AND PERFORMANCE METRICS?

Fig. 8 illustrates a heat map for Pearson correlation between utility functions and performance metrics. We investigate each case separately to determine whether scaling up the dataset affects results. For a complete analysis, we evaluate all policies reported in Figs. 3, 7d and 7f, comparing the progression of trained agents; thus, our analysis includes agents with varying capabilities. Within utility function categories, there is a trend of positive correlation, except for realism, unsurprisingly. Imagine an RL agent taking a turn at an intersection, turning earlier/later than the logged trajectory, causing high $U^{\text{GC-ADE}}$ yet low $U^{\text{Act-MAE}}$, as, apart from the moment when the agent takes a turn, it will act similarly. Note that identical/divergent sequences of actions also lead to the same/distinct trajectories, respectively; hence, there is no clear correlation. In contrast, regret-based functions tend to be positively correlated, since, though in different ways, they all approximate regret. Interestingly, the correlation between these utilities increases monotonically as the training dataset grows, possibly because improved value estimation also improves TD-error estimation for $U^{\text{AMGAE}}$ and $U^{\text{PVL}}$, resulting in better performance in case 2 than in case 1. Finally, success-based functions measure the variance of similar statistics; hence, they have a high positive correlation.

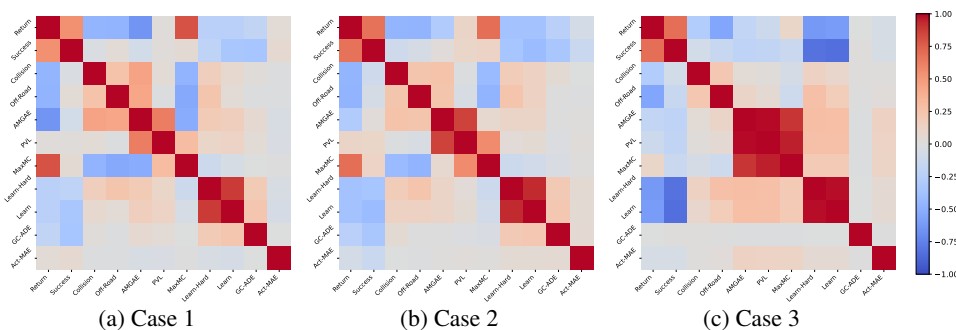

Figure 8: Pearson correlation between utility functions and performance metrics, i.e, success, collision, and off-road rates: Results from training in **(a)** 1,000 **(b)** 10,000, and **(c)** 80,000 scenarios.

Across categories, though not high, there is a positive correlation between regret- and success-based utility functions because, in general, high regret scenarios have high success variance. Such cases yield high TD errors, as the agent can achieve a high return but has a low estimated value because it is not optimal yet. Realism-based functions are not necessarily correlated with the rest, except for the small correlation in case 3. The reason is that realism is not equivalent to optimality with respect to a reward function, which, in GPUDRIVE, incentivizes reaching the goal as quickly as possible while avoiding collisions and going off-road. An RL agent can behave optimally in terms of such a basic reward function, yet also not realistically, as the reward function does not address attributes such as comfort, staying within lanes, or going under the speed limit.

Finally, we investigate how utility functions correlate with performance metrics. $U^{\text{AMGAE}}$ correlates with collision/off-road rates, with the highest in case 1, likely leading to $U^{\text{AMGAE}}$ underperforming, as it prioritizes scenarios with crashes/off-road events. $U^{\text{PVL}}$ does not correlate with any performance metrics. In contrast to $U^{\text{AMGAE}}$, $U^{\text{MaxMC}}$ shows a positive correlation with returns. Although this leads to $U^{\text{MaxMC}}$ outperforming most PLR variants in case 1, as the correlation decreases, its performance degrades as well. Success-based functions have a negative correlation with return and success, and a positive correlation with collision/off-road rates. This is possibly because high variance in success occurs when the agent collects low returns. Realism-based functions do not correlate with any performance metrics, in general, likely because, as aforementioned, realistic (unrealistic) behavior does not necessarily correspond to optimal (suboptimal) policy.

## 6 CONCLUSION

In this work, we introduce CL4AD, the first integration of CL into batched AD simulators. CL4AD frames scenario selection as a UED problem, enabling adaptive prioritization of traffic scenarios via a well-known method, PLR (Jiang et al., 2021b), combined with utility functions that measure the regret, success, and realism of the trained agent's behavior. We conduct extensive large-scale experiments by integrating CL4AD into GPUDRIVE, an open-source batched AD simulator. Empirically, curriculum learning achieves 99% goal-completion in test scenarios up to 77% faster than domain randomization, i.e., the default scenario sampling technique, when trained with datasets ranging from 1,000 to 80,000 traffic scenarios. CL4AD further demonstrates that, CL reduces wall-clock time to reach the same success rate by 67% under limited compute resources, as well.

**Limitations and future work.** CL4AD evidences that CL scales up to the high-throughput of batched AD simulators. However, CL4AD is currently limited to an implementation of PLR and requires access to a real self-driving dataset as a source of traffic scenarios for sampling, e.g., WOMD, since GPUDRIVE operates on pre-defined scenarios. To address these limitations, future work will explore UED methods such as ACCEL (Parker-Holder et al., 2022), which randomly mutates prioritized scenarios, hence increasing scenario diversity for training robust policies. In addition, synthetic scenario generation tools, e.g., Scenario Dreamer (Rowe et al., 2025), can enable CL4AD to further accelerate training and improve the robustness and generalization capabilities of trained agents by creating safety-critical or out-of-distribution scenarios that the agent struggles with.

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

## A    LANGUAGE MODEL USE STATEMENT

This work utilized language models for brief editing to enhance clarity and conciseness.

## B    REPRODUCIBILITY STATEMENT

For all the hyperparameters and detailed settings of the experiments, please refer to Appendix E. We provide the implementation of CL4AD in GPUDRIVE in an anonymized repository: https://anonymous.4open.science/r/gpudrive-37D3/README.md. We also include instructions to install the required software, run experiments, and reproduce results in a README file. The traffic scenarios in WOMD can be accessed on the repository for GPUDRIVE: https://github.com/Emerge-Lab/gpudrive.

## C  NOMENCLATURE

| | |
|---|---|
| $\mathcal{G}$ | POSG |
| $\mathcal{N}, \mathrm{N}$ | Set of agents, number of agents ($|\mathcal{N}| = \mathrm{N}$) in POSG |
| $\mathcal{S}, \mathcal{A}, \mathcal{O}$ | State, action and observation spaces in POSG |
| $\mathbf{s}, \mathbf{a}, \mathbf{o}, r$ | State, action, observation, and reward in POSG |
| $T, Z, R$ | Transition, observation, and reward functions in POSG |
| $I$ | Initial state distribution in POSG |
| $\gamma, \mathrm{H}$ | Discount factor and horizon in POSG |
| $\Theta, \theta$ | Set of scenarios and scenarios, i.e., $\theta \in \Theta$, in UPOSG |
| $\mathcal{G}^\Theta$ | UPOSG |
| $\mathcal{N}^\Theta, \mathrm{N}$ | Set of agents, number of agents ($|\mathcal{N}^\Theta| = \mathrm{N}$) in UPOSG |
| $\mathcal{S}, \mathcal{A}^\Theta, \mathcal{O}^\Theta$ | State, action and observation spaces in UPOSG |
| $\mathbf{s}, \mathbf{a}, \mathbf{o}, r$ | State, action, observation, and reward in UPOSG |
| $T^\Theta, Z^\Theta, R^\Theta$ | Transition, observation, and reward functions in UPOSG |
| $I^\Theta$ | Initial state distribution in UPOSG |
| $\gamma, \mathrm{H}$ | Discount factor and horizon in UPOSG |
| $\mathbf{x}$ | Position of an agent in a scenario in UPOSG |
| $\mathrm{M}$ | Number of scenarios in a UPOSG, i.e., $|\Theta| = \mathrm{M}$ |
| $\mathcal{S}^\theta_{\mathrm{Goal}}$ | Goal states in a UPOSG |
| $\Lambda$ | Level generator for UED |
| $\Pi$ | Policy space in UED |
| $\Delta(\Theta)$ | Distribution over levels in UED |
| $U, \mathrm{C}$ | Utility function in UED, constant utility in UED |
| $\mathcal{B}, \mathbb{P}_{\mathrm{replay}}$ | Replay buffer and distribution in PLR |
| $\mathbb{P}_{\mathrm{utility}}, \mathbb{P}_{\mathrm{staleness}}$ | Score and staleness distribution in PLR |
| $l$ | Scenario sampling iteration in PLR |
| $\rho, \beta, d, \mathrm{B}^{\mathrm{max}}$ | Staleness coefficient, score temperature, replay rate, max replay buffer size |
| $\pi, V$ | Policy, value function |
| $\lambda, \delta$ | GAE discount factor, TD error |
| $\mathrm{R}^{\theta,n}_{\mathrm{max}}$ | Maximum return of an agent |
| $p$ | Success rate |
| $\phi$ | Trainable policy parameter |
| $\mathrm{T}^{\mathrm{train}}, \mathrm{T}^{\mathrm{sce}}, \mathrm{T}^{\mathrm{pol}}$ | Number of interactions for training, sampling scenarios, and updating policy |
| $\mathrm{W}$ | Number of concurrent worlds |
| $\mathcal{D}$ | Experience buffer |
| $\Phi()$ | RL algorithm of choice to update policy |
| $e$ | End of episode flag |
| $\tau$ | Rollout |

**Algorithm 2** SAMPLEFROMCURRICULUM()

**Input**: Replay buffer $\mathcal{B}$, set of training scenarios $\Theta^{\text{train}}$, sampling iteration $l$
**Parameters**: Replay rate $d$, staleness $\rho$, temperature $\beta$, max buffer size $B^{\text{max}}$, number of worlds W
**Output**: Sampled scenarios $(\theta_w)_{w=1}^{W}$, and buffer $\mathcal{B}$ with updated staleness
1: $\mathcal{B} \leftarrow$ DISCARDLOWESTRANKINGSCENARIOS$(\mathcal{B}, B^{\text{max}})$
2: **if** $|\mathcal{B}| \equiv 0$ **or** (Bernoulli$(d) \equiv 0$ **and** $|\Theta^{\text{train}} - \mathcal{B}^{\text{scenario}}| > 0$) **then**
3:     $\mathbb{P}_{\text{sample}} \leftarrow$ Uniform$(\Theta^{\text{train}} - \mathcal{B}^{\text{scenario}})$           ▷ *Uniformly randomly sample scenarios*
4: **else**
5:     $\mathbb{P}_{\text{sample}} \leftarrow \mathbb{P}_{\text{replay}}$                     ▷ *Replay scenarios based on $\mathbb{P}_{replay}$*
6: $(\theta_w)_{w=1}^{W} \leftarrow$ Sample$(\mathbb{P}_{\text{sample}}, W)$        ▷ *Sample W-many scenarios based on $\mathbb{P}_{sample}$*
7: $\mathcal{B}^{\text{scenario}} \leftarrow \mathcal{B}^{\text{scenario}} \cup (\theta_w)_{w=1}^{W}$           ▷ *Update scenarios in the replay buffer*
8: $l_{\theta_w} \leftarrow l, \forall w \in [W]$         ▷ *Update sampling iteration for staleness distribution*
9: $\tau_{\theta_w} \leftarrow (), \forall w \in [W]$                          ▷ *Reset the rollout*

**Algorithm 3** UPDATECURRICULUM()

**Input**: Interaction set $\mathcal{D}_t$, utility function $U$, replay buffer $\mathcal{B}$
**Output**: Updated replay buffer $\mathcal{B}$
1: **for** $w \in [W]$ **do**
2:    **if** $e_{n,w}$ is True $\forall n \in [N_{\theta_w}]$ **then**
3:       score$_{\theta_w, t} \leftarrow U(\tau_{\theta_w})$         ▷ *Compute utility score for terminated episode*
4:       score$_{\theta_w} \leftarrow$ MovingAverage(score$_{\theta_w}$, score$_{\theta_w, t}$)     ▷ *Update the score in the buffer*
5:       $\tau_{\theta_w} \leftarrow ()$                                ▷ *Reset the rollout*
6:    **else**
7:       $\tau_{\theta_w} \leftarrow \tau_{\theta_w} \cup \{\mathbf{o}_{n,w}, \mathbf{a}_{n,w}, \mathbf{o}'_{n,w}, r_{n,w}, e_{n,w}\}_{n \in [N_{\theta_w}]}$ ▷ *Update the rollout with new interactions*

# D    DETAILS OF CL4AD

In this section, we provide a more detailed look into how CL4AD works to support the material in Section 4. Algorithm 2 is a pseudocode for how CL4AD samples new scenarios during training via PLR. First, CL4AD removes scenarios with ranking lower than $B^{\text{max}}$ in the buffer (Line 1), where $B^{\text{max}}$ is the maximum size of $\mathcal{B}$ for sampling. If the buffer size is smaller than or equal to $B^{\text{max}}$, then no scenario is removed. Then, it determines whether to sample traffic scenarios from the replay buffer. If the replay buffer is empty, or the random replay decision is False, conditioned on the fact that there are still unseen scenarios, then CL4AD uniformly randomly samples unseen scenarios from the training dataset. Otherwise, it uses the replay distribution $\mathbb{P}_{\text{replay}}$ to sample from the replay buffer $\mathcal{B}$ (lines 2-6). Then, CL4AD updates the scenarios in the buffer with the newly sampled ones and sets their corresponding last sampling iteration to the current one for staleness computation later on (lines 7-9). Algorithm 3 is a pseudocode for how CL4AD updates the buffer. CL4AD goes through every world and checks whether an episode has terminated. If so, it computes the utility of that episode based on the rollout that CL4AD has kept track of. Then, this score is used to update the score in the buffer via moving average, and finally, the rollout is reset for a new episode to save memory. If the episode continues, CL4AD updates the rollouts with the latest interactions.

# E    EXPERIMENTAL DETAILS

In this section, we describe the process of hyperparameter selection for our experiments.

## E.1    SIMULATION SET-UP

Our integration of CL4AD into GPUDRIVE follows the simulation set-up in Kazemkhani et al. (2025), where the simulator ignores collisions and going off-road, i.e., they do not lead to episode termination; the observation of a vehicle is its bird-eye-view of a radius of 50m; non-vehicle objects are omitted; a goal is considered to be achieved if an agent is in its proximity by 2m; the action consists of two discrete random variables for steering and acceleration inputs, divided into evenly spaced grids, 13 and 7, respectively; maximum number of controlled agents in a scenario is 64;

Table 1: Self-play PPO Hyperparameters

| Parameter | Case 1,2,3 | Ablation |
|---|---|---|
| total_timesteps $T^{\text{train}}$ | $2,000,000,000$ | $1,000,000,000$ |
| num_worlds W | 800 | 100 |
| batch_size $T^{\text{pol}}$ | $524,288$ | $131,072$ |
| minibatch_size | $16,384$ | $8,192$ |
| learning_rate | 0.0003 | 0.0003 |
| anneal_lr | false | false |
| gamma $\gamma$ | 0.99 | 0.99 |
| gae_gamma $\lambda$ | 0.95 | 0.95 |
| update_epochs | 2 | 4 |
| norm_adv | true | true |
| clip_coef | 0.2 | 0.2 |
| clip_vloss | false | false |
| vf_clip_coef | 0.2 | 0.2 |
| ent_coef | 0.0001 | 0.0001 |
| vf_coef | 0.5 | 0.3 |
| max_grad_norm | 0.5 | 0.5 |
| target_kl | null | null |
| collision_weight | $-0.75$ | $-0.75$ |
| off_road_weight | $-0.75$ | $-0.75$ |
| goal_achieved_weight | 1.0 | 1.0 |

the agents only observe the current time step; and the episode takes 90 timesteps, amounting to 9 seconds, at most. For more details, we refer the reader to the default PufferLib configuration (see environment section) in the repository published by Kazemkhani et al. (2025).

### E.2 SELF-PLAY PPO TRAINING

Table 1 lists the hyperparameters for self-play PPO training in cases 1, 2, and 3, as well as the ablation study. As the ablation study investigates limited compute resources, i.e., the use of fewer worlds and lower batch sizes, we essentially set them according to the hyperparameters in Kazemkhani et al. (2025), where the number of worlds W = 50. In comparison, cases 1, 2, and 3 studies a larger scale in terms of throughput, hence utilize significantly more concurrent worlds and a larger experience buffer. As a result, their hyperparameters come from Cornelisse et al. (2025), which focuses on a similar scale. The weights for collision/off-road penalties and goal completion rewards also come from Cornelisse et al. (2025). For cases 1 and 2, as well as the ablation study, the experiments are over three independent runs, utilizing seeds 42, 12, and 67. Case 3 uses seeds 42 and 12. The network architecture also follows the settings in Cornelisse et al. (2025).

### E.3 SCENARIO SAMPLING DETAILS

Table 2 demonstrates the hyperparameters used for the experiments we report in cases 1, 2, 3, and the ablation study. The search space for PLR hyperparameters is as follows: staleness coefficient $\rho \in \{0.1, 0.2\}$ and score temperature $\beta \in \{2, 4\}$, based mainly on Jiang et al. (2021b). We first conduct a grid search in Case 1, where we train agents using all score functions on three independent runs for one billion interactions. Then we select the pair that yields the highest success rate, the fastest at test-time. Jiang et al. (2021a) suggests a lower temperature; however, our experiments indicate that a higher temperature, especially considering the size of the training dataset, is more performant in large-scale training. Case 3 and the ablation study also utilize these hyperparameters. In case 2, we find that a higher temperature yields better results. We set the replay buffer size to the size of the training dataset, and sample scenarios every $2,000,000$ interactions.

Table 2: Case 1: PLR Hyperparameters

| | Utility Function | $d$ | $\beta$ | $\rho$ |
|---|---|---|---|---|
| Case 1 | $U^{\text{Act-MAE}}$ | 0.5 | 2 | 0.3 |
| | $U^{\text{AMGAE}}$ | 0.5 | 4 | 0.3 |
| | $U^{\text{GC-ADE}}$ | 0.5 | 4 | 0.3 |
| | $U^{\text{Learn}}$ | 0.5 | 4 | 0.1 |
| | $U^{\text{Learn-hard}}$ | 0.5 | 2 | 0.1 |
| | $U^{\text{MaxMC}}$ | 0.5 | 2 | 0.3 |
| | $U^{\text{PVL}}$ | 0.5 | 4 | 0.3 |
| Case 2 | $U^{\text{Act-MAE}}$ | 0.5 | 4 | 0.3 |
| | $U^{\text{Learn}}$ | 0.5 | 4 | 0.1 |
| | $U^{\text{MaxMC}}$ | 0.5 | 4 | 0.3 |
| Case 3 | $U^{\text{Learn}}$ | 0.5 | 4 | 0.1 |
| | $U^{\text{MaxMC}}$ | 0.5 | 2 | 0.3 |
| Ablation | $U^{\text{Act-MAE}}$ | 0.5 | 2 | 0.3 |
| | $U^{\text{Learn}}$ | 0.5 | 4 | 0.1 |
| | $U^{\text{MaxMC}}$ | 0.5 | 2 | 0.3 |

# F  COMPUTATIONAL RESOURCES

We run our experiments in cases 1, 2, and 3 on an NVIDIA H200, which has 141 GB of GPU memory. One training run, which amounts to 2 billion steps and approximately 3,800 policy updates, takes around 60 hours. For the ablation study, we train agents on NVIDIA RTX A5000, which has a GPU memory of 24GB, for a billion interactions, which takes over 110 hours.

# G  DETAILED RESULTS

## G.1  QUANTITATIVE RESULTS

Figures 9, 10, 11, and 12 demonstrate the progression of trained agents in cases 1, 2, and 3, as well as the ablation study, respectively. These figures provide details on the progression of performance, regret, realism, and learnability when agents are evaluated in the training and test partitions of their respective experiments. Regret, learnability, and realism in the training partition highlight how automated curricula impact training. In most cases, we observe that PLR variants are significantly faster than DR at achieving low utility scores in these metrics, indicating that they obtain more performant and realistic policies more quickly. The performance progression, when evaluated on the training partition, leads to a similar observation as well. Progression in test scenarios demonstrates the generalization capabilities of these trained agents, as these scenarios were not encountered during training. Overall, we observe that PLR variants are again quickly becoming more capable at generalization or becoming robust and reliable faster than agents trained via DR.

## G.2  QUALITATIVE RESULTS

Figures 13, 14, 15, 16, 17, and 18 illustrate the $\mathbb{P}_{\text{replay}}$ progression of PLR in case 1. Here we omit $U^{\text{MaxMC}}$, as we provide its illustration in the main document. The utility functions with a high score temperature, i.e., $\beta = 4$, as opposed to $\beta = 2$, lead to a more uniform replay distribution (see Figures 14, 15, 17, 18 for $U^{\text{AMGAE}}$, $U^{\text{GC-ADE}}$, $U^{\text{Learn}}$, and $U^{\text{PVL}}$, respectively). As the score

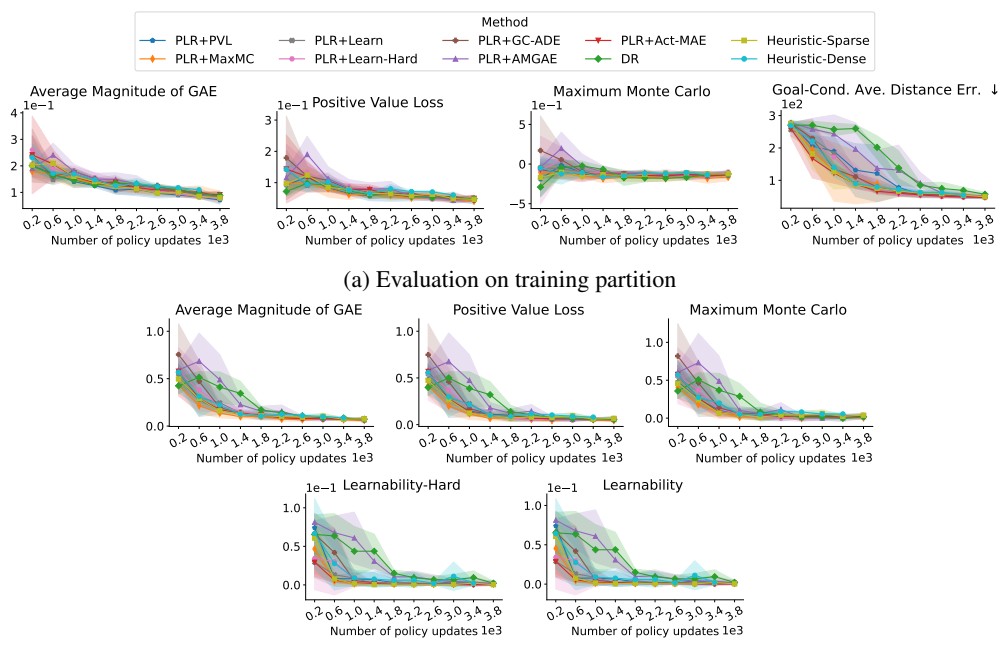

(a) Evaluation on training partition

(b) Evaluation on test partition

Figure 9: Case 1: Regret ($U^{\text{AMGAE}}$, $U^{\text{PVL}}$, $U^{\text{MaxMC}}$), realism ($U^{\text{GC-ADE}}$), and learnability ($U^{\text{Learn}}$, $U^{\text{Learn-hard}}$), progression during training with 1000 scenarios from WOMD: We evaluate in **(a)** training partition, and **(b)** 150 test scenarios. Bold markers indicate the mean, whereas the shaded area covers one standard deviation around it across three independent training runs.

temperature decreases, the impact of the ranking on the replay distribution also decreases. Furthermore, we observe that certain utility functions result in significant changes in the replay distribution throughout training, specifically when visualized with respect to the number of controlled agents in scenarios (see Figures 13 and 16 for $U^{\text{Act-MAE}}$ and $U^{\text{Learn-hard}}$, respectively). The reason behind such changes may be the use of a lower score temperature, which allows the ranking to impact the replay distribution more drastically.

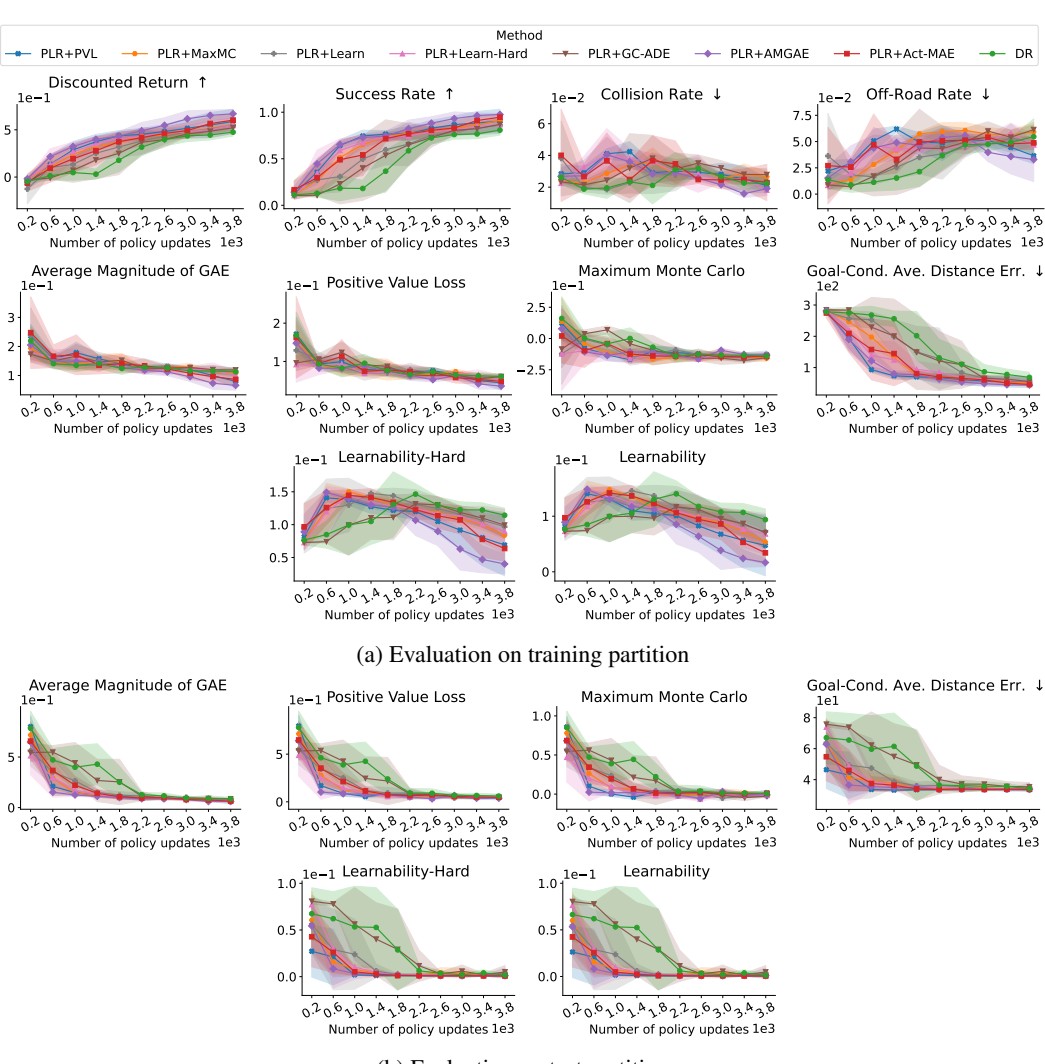

Figure 10: Case 2: Performance, Regret ($U^{\text{AMGAE}}$, $U^{\text{PVL}}$, $U^{\text{MaxMC}}$), realism ($U^{\text{GC-ADE}}$), and learnability ($U^{\text{Learn}}$, $U^{\text{Learn-hard}}$), progression during training with 10,000 scenarios from WOMD: We evaluate in **(a)** training partition, and **(b)** 10,000 test scenarios. Bold markers indicate the mean, whereas the shaded area covers one standard deviation around it across three training runs.

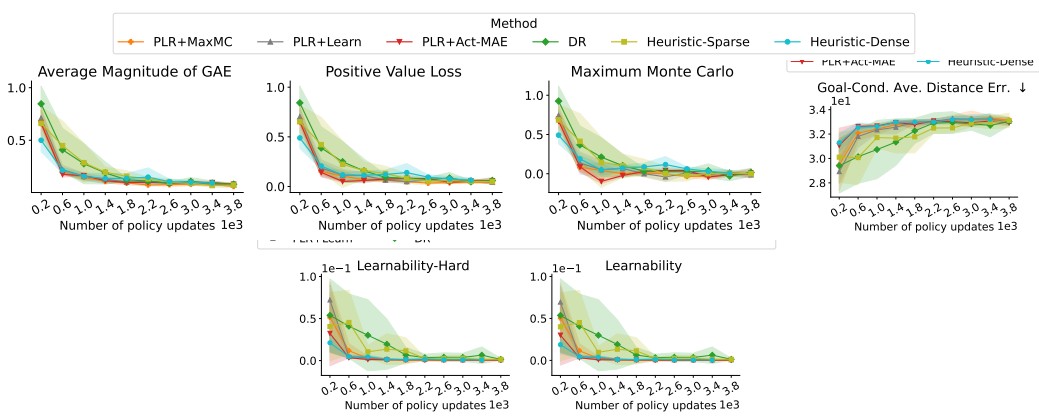

Figure 11: Case 3: Regret ($U^{\mathrm{AMGAE}}$, $U^{\mathrm{PVL}}$, $U^{\mathrm{MaxMC}}$), realism ($U^{\mathrm{GC\text{-}ADE}}$), and learnability ($U^{\mathrm{Learn}}$, $U^{\mathrm{Learn\text{-}hard}}$), progression during training with 80,000 scenarios from WOMD: We evaluate in 10,000 test scenarios. Bold markers indicate the mean, whereas the shaded area covers one standard deviation around it across two independent training runs.

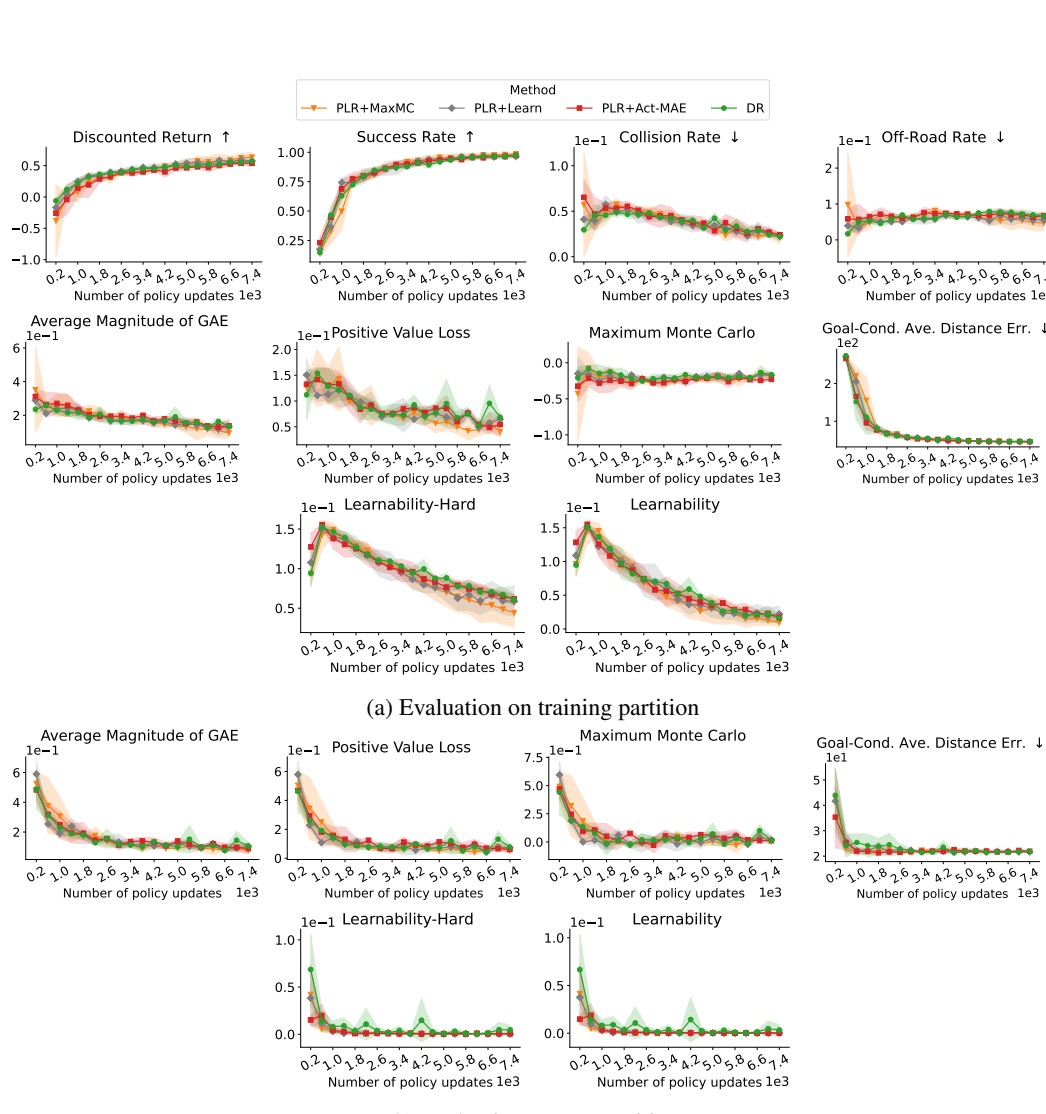

Figure 12: Ablation: Performance, regret ($U^{\mathrm{AMGAE}}$, $U^{\mathrm{PVL}}$, $U^{\mathrm{MaxMC}}$), realism ($U^{\mathrm{GC\text{-}ADE}}$), and learnability ($U^{\mathrm{Learn}}$, $U^{\mathrm{Learn\text{-}hard}}$), progression during training for our ablation study on compute resources: We evaluate in **(a)** training partition, and **(b)** 150 test scenarios. Bold markers indicate the mean, whereas the shaded area covers one standard deviation around it across three training runs.

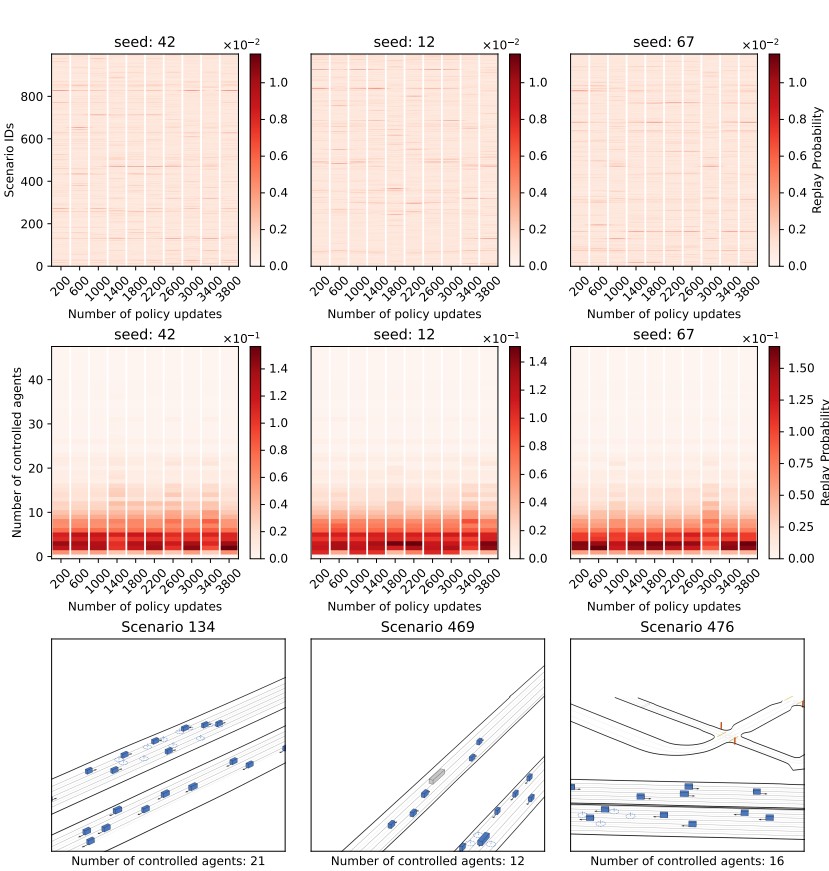

Figure 13: $\mathbb{P}_{\text{replay}}$ progression of PLR combined with $U^{\text{Act-MAE}}$ in mini WOMD: We illustrate **(top)** the evolution of $\mathbb{P}_{\text{replay}}$, where darker line segments indicate scenarios with higher replay likelihood, **(middle)** a version of replay distribution under categorization with respect to the number of controlled agents in scenarios, and **(bottom)** we exemplify three scenarios that appear frequently.

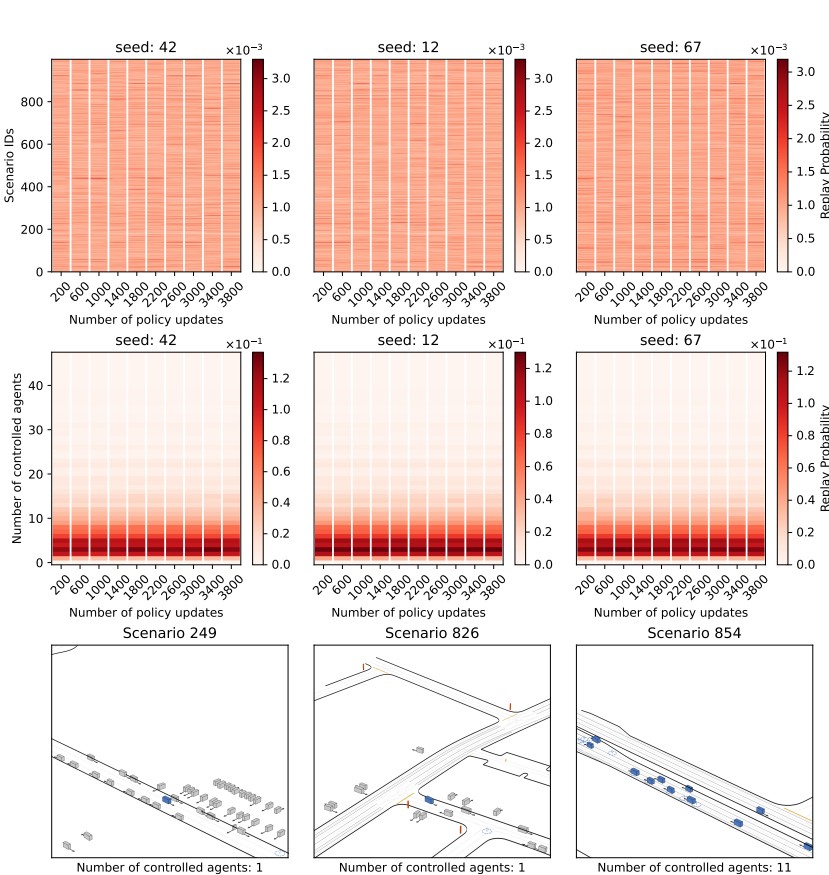

Figure 14: $\mathbb{P}_{\text{replay}}$ progression of PLR combined with $U^{\text{AMGAE}}$ in mini WOMD: We illustrate **(top)** the evolution of $\mathbb{P}_{\text{replay}}$, where darker line segments indicate scenarios with higher replay likelihood, **(middle)** a version of replay distribution under categorization with respect to the number of controlled agents in scenarios, and **(bottom)** we exemplify three scenarios that appear frequently.

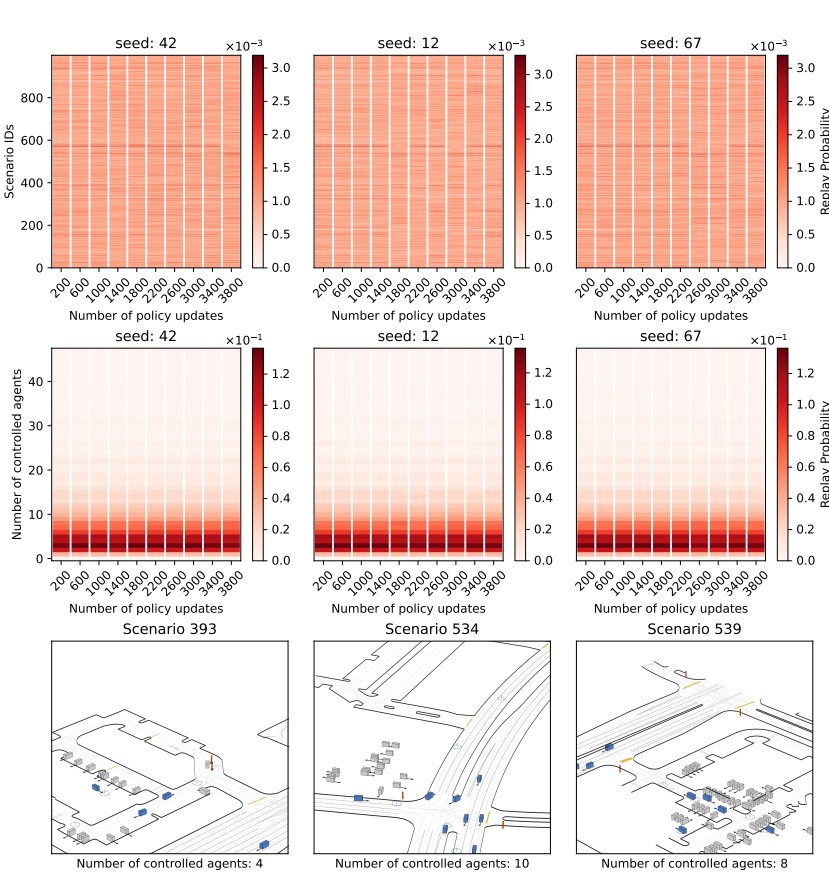

Figure 15: $\mathbb{P}_{\text{replay}}$ progression of PLR combined with $U^{\text{GC-ADE}}$ in mini WOMD: We illustrate **(top)** the evolution of $\mathbb{P}_{\text{replay}}$, where darker line segments indicate scenarios with higher replay likelihood, **(middle)** a version of replay distribution under categorization with respect to the number of controlled agents in scenarios, and **(bottom)** we exemplify three scenarios that appear frequently.

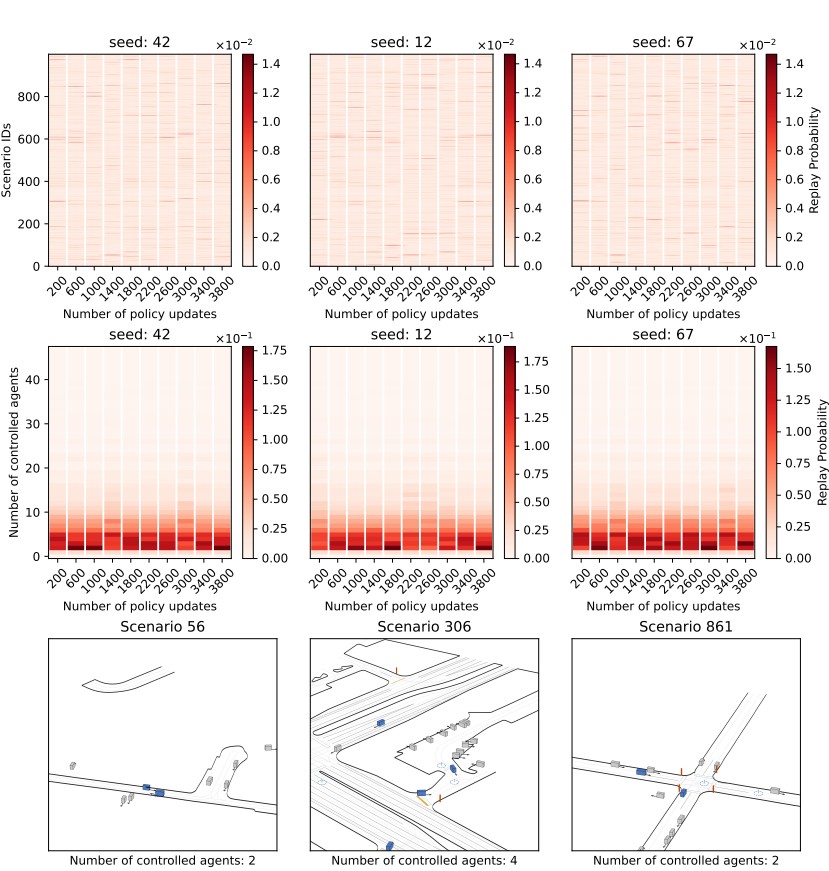

Figure 16: $\mathbb{P}_{\text{replay}}$ progression of PLR combined with $U^{\text{Learn-hard}}$ in mini WOMD: We illustrate **(top)** the evolution of $\mathbb{P}_{\text{replay}}$, where darker line segments indicate scenarios with higher replay likelihood, **(middle)** a version of replay distribution under categorization with respect to the number of controlled agents in scenarios, and **(bottom)** we exemplify three scenarios that appear frequently.

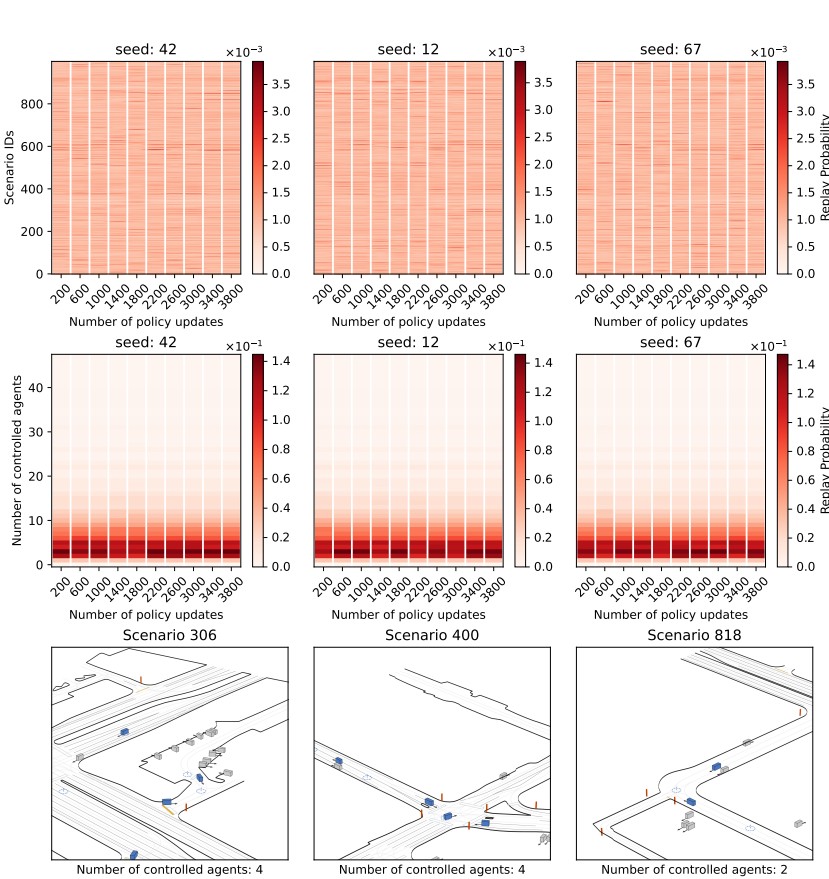

Figure 17: $\mathbb{P}_{\mathrm{replay}}$ progression of PLR combined with $U^{\mathrm{Learn}}$ in mini WOMD: We illustrate **(top)** the evolution of $\mathbb{P}_{\mathrm{replay}}$, where darker line segments indicate scenarios with higher replay likelihood, **(middle)** a version of replay distribution under categorization with respect to the number of controlled agents in scenarios, and **(bottom)** we exemplify three scenarios that appear frequently.

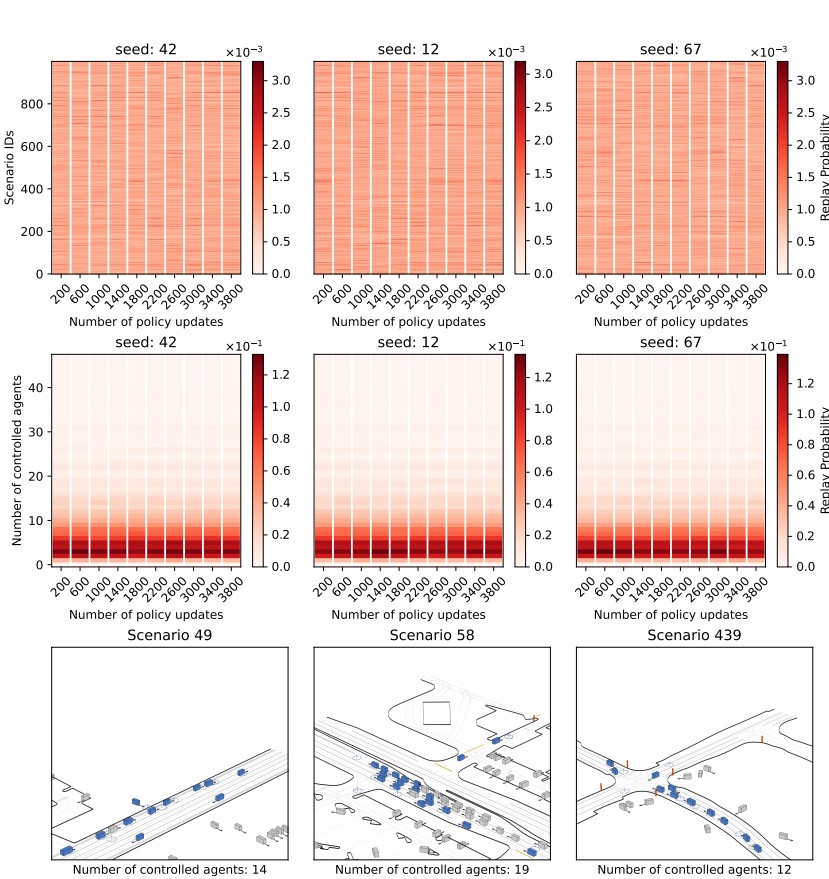

Figure 18: $\mathbb{P}_{\mathrm{replay}}$ progression of PLR combined with $U^{\mathrm{PVL}}$ in mini WOMD: We illustrate **(top)** the evolution of $\mathbb{P}_{\mathrm{replay}}$, where darker line segments indicate scenarios with higher replay likelihood, **(middle)** a version of replay distribution under categorization with respect to the number of controlled agents in scenarios, and **(bottom)** we exemplify three scenarios that appear frequently.

