# OpenReview forum: "Scaling Curriculum Learning for Autonomous Driving"
_ICLR.cc/2026/Conference — Submitted to ICLR 2026_

### Official Review · Reviewer_rqJS · 2025-10-15

**Soundness:** 3
**Presentation:** 3
**Contribution:** 3
**Rating:** 6
**Confidence:** 4

**Summary:**

The paper presents CL4AD, a scalable curriculum learning framework for batched autonomous driving simulators such as GPUDRIVE. It integrates unsupervised environment design methods to adaptively prioritize driving scenarios based on regret, success, and realism. The authors design new utility functions tailored for autonomous driving and show that CL4AD improves sample efficiency and convergence speed compared to uniform domain randomization.

**Strengths:**

1. Addresses an important and timely problem: improving sample efficiency in large-scale driving RL through adaptive scenario selection.
2. Provides a well-engineered integration of UED methods into high-throughput simulators, with clear algorithmic formulation and open-source implementation.
3. Large-scale experiments convincingly show faster convergence and reduced compute cost compared to domain randomization, backed by detailed ablations on compute limits and dataset size.

**Weaknesses:**

1. The novelty is moderate: the method mainly adapts existing PLR-based UED techniques to the AD domain, with limited conceptual innovation beyond utility function design.
2. The proposed realism-based utility functions are interesting but insufficiently justified. It is unclear whether they consistently outperform regret-based metrics or simply correlate with progress.

**Questions:**

1. Could the authors quantify the computational overhead introduced by CL4AD compared to standard domain randomization, especially in terms of GPU time and memory?
2. How sensitive is the performance to the choice and scaling of different utility functions? Are some functions redundant or correlated?

---

> ### Author Response · Authors · 2025-11-27
>
> We want to thank the reviewer for their fair criticism and helpful questions, which helped us improve our work. Below, we first address their comments and then their questions.
>
> **W1**: We would like to clarify the contribution of our work, as we consider it to be more than a mere adaptation of UED in the AD domain.
>
> The contribution of CL4AD is in the form of
> 1) integration of curriculum learning into a batched AD simulator for the first time,
> 2) scaling CL for tens of controllable agents in a single scenario, hundreds of concurrent worlds, tens of thousands of training scenarios, and billions of training iterations, and
> 3) proposing novel utility functions, evaluating new and existing ones across multiple scaling dimensions, and analyzing their correlation with each other and performance metrics.
>
> We think these are valuable contributions, as automated curricula have been studied only a few times in the autonomous driving context and have not been attempted at scale. Therefore, our work paves the way for future research on curriculum learning algorithms in industry-scale simulators.
>
> **W2**: To address the reviewer's comments on justification, we have provided a new, comprehensive analysis of utility functions, discussing their correlation with performance metrics in Section 5.5. We thank the reviewer for pointing out this issue, as we believe this new subsection sheds light on why certain utility functions perform better than others and under what conditions.
>
> About whether proposed realism-based utility functions outperform existing ones, we do not have such a claim in our paper. We demonstrate that realism-based functions, commonly used to evaluate AD policies, can effectively prioritize scenarios to accelerate learning. Especially, Act-MAE can consistently compete with existing utility functions. Note that realism-based utility functions measure likeness to human/logged behavior, which is only one solution to AD, and may not even be optimal with respect to the reward function. Section 5.5. discusses these issues in more detail.
>
> **Q1**: We analyzed time and memory requirements in an ablation study using limited compute resources. Below, we compare the time taken to take a step in all worlds simultaneously, to update the PLR buffer, sample via PLR, and update the policy:
> - Taking a step in GPUDrive: 0.105 +- 0.015 seconds
> - Updating PLR buffer: 0.0003 +- 0.00005 seconds
> - Sampling via PLR: 0.95 +- 0.03 seconds
> - Updating policy: 3.1 +- 0.4 seconds
>
> Note that taking a step and updating the buffer occur with the same frequency, and the latter is 350 times faster. In comparison, sampling via PLR and updating policy occur once every two million and one hundred thousand interactions, respectively. In short, CL4AD has a minor impact on training time, as it is designed to process all worlds and agents in parallel.
>
> In terms of memory, CL4AD keeps track of all agents until their current episode terminates. As a result, it has dynamic memory whose size changes with the total number of controllable agents in the current world. For example, in the ablation study, the memory size ranges from 50 to 80 thousand (100 worlds * 91 timesteps per episode * 6-9 agents on average). In comparison to the PPO's experience buffer (with a size greater than 130,000) for policy updates, which also keeps track of high-dimensional observations, PLR only records scalar values, e.g., rewards, values, success/collision/off-road signals.
>
> **Q2**: Please see the new discussion in Section 5.5, which we also summarize in our global response.

---

> > ### Comment · Reviewer_rqJS · 2025-11-28
> >
> > I thank the authors for their detailed response, particularly the clarifications on computational overhead and the additional analysis in Section 5.5. After reviewing the rebuttal and the other reviews, while I acknowledge the engineering effort involved in scaling, I still consider this work to be more of an application paper than a fundamental contribution to RL methods. Furthermore, the lack of evaluation on public benchmarks, such as WOSAC, limits the empirical validation. Therefore, I will maintain my original score.

---

> ### Author Response · Authors · 2025-11-28
>
> We thank the reviewer for their timely response. We want to clarify that we have submitted our work under the primary area **'applications to robotics, autonomy, planning'** since our main contribution is being the first to integrate, scale, and analyze curriculum learning in the context of batched AD simulators.
>
> In addition, the Waymo Open Motion Dataset (WOMD) [1], which we used at different scales in our work to investigate the benefits of curriculum learning in GPUDrive, is a widely accepted public benchmark that has been the focus of many research papers [2-4].
>
> **References:**
> 1) Ettinger, S., Cheng, S., Caine, B., Liu, C., Zhao, H., Pradhan, S., ... & Anguelov, D. (2021). Large scale interactive motion forecasting for autonomous driving: The waymo open motion dataset. In Proceedings of the IEEE/CVF international conference on computer vision (pp. 9710-9719).
> 2) Rowe, L., Girgis, R., Gosselin, A., Carrez, B., Golemo, F., Heide, F., ... & Pal, C. (2025, January). CtRL-Sim: Reactive and Controllable Driving Agents with Offline Reinforcement Learning. In Conference on Robot Learning (pp. 3600-3621). PMLR.
> 3) Son, S., Zheng, L., Clipp, B., Greenwell, C., Philip, S., & Lin, M. C. (2025, May). Gradient-based Trajectory Optimization with Parallelized Differentiable Traffic Simulation. In 2025 IEEE International Conference on Robotics and Automation (ICRA) (pp. 14497-14504). IEEE.
> 4) Kazemkhani, S., Pandya, A., Cornelisse, D., Shacklett, B., & Vinitsky, E. (2025, May) GPUDrive: Data-driven, multi-agent driving simulation at 1 million FPS. In The Thirteenth International Conference on Learning Representations.

---

### Official Review · Reviewer_oqQS · 2025-10-20

**Soundness:** 3
**Presentation:** 3
**Contribution:** 2
**Rating:** 4
**Confidence:** 4

**Summary:**

Based on GPUDrive, this paper investigates how curriculum learning (CL) can accelerate training driving agents in the self-play setting. Besides 3 existing CL unitility functions, this work proposed 3 new utility functions and benchmarked these methods on GPUDrive and compared the results with uniform sampling. The results show that introducing CL into the training can indeed largely speed up the policy training in terms of success rate and other metrics.

**Strengths:**

1. This is the first work that integrates CL into batched autonomous driving simulators by framing scenario selection as an unsupervised environment design problem.
2. The effectiveness of CL method is confirmed in this setting
3. The paper is easy to read

**Weaknesses:**

1. This work benchmarks existing Unsupervised Evironment Design (UED) methods, especially PLR-based ones, under the bacthed driving simulator setting. Though three new utility functions used in PLR are proposed, they didn't outperform those proposed in previous research (PLR-learn and PLR-MaxMC). In addition, a large part of the method section is to introduce previous works, making this paper looks like a benchmark paper. Considering that CL helps training is a common sense, this paper lack of novelty on both the method and conclusion.

2. As this paper is engineering-oriented, one good direction to revise is to tune/benchmark the agents on widely-used benchmarks like WOSAC and nuPlan like what the Giga flow did. This can bring real-world impact. It would be better to see how CL can help self-replay agents achieve SOTA on those benchmarks in terms of performance and training speed. The outcome will add a lot of values to this paper, making it an new paradigm for addressing the agent simulation & planning. In addition, I believe new problems will appear in this process, which may even inspire authors to design new UED methods to improve the novelty of this paper.

3. Several related works are missing:
- Discovering General Reinforcement Learning Algorithms with Adversarial Environment Design by Matthew T. Jackson et al., which combines the idea of meta-learning and UED.
- ScenarioNet: Open-Source Platform for Large-Scale Traffic Scenario Simulation and Modeling By Quanyi Li et al., which also applied CL into the training of driving agents.

**Questions:**

N/A

---

> ### Author Response · Authors · 2025-11-27
>
> We thank the reviewer for taking the time to review our work and for their suggestions to improve the related work section. Below, we address their comments.
>
> **W1**: Indeed, the reviewer is correct that the contribution of CL4AD is in the form of
> 1) integration of curriculum learning into a batched AD simulator for the first time,
> 2) scaling CL for tens of controllable agents in a single scenario, hundreds of concurrent worlds, tens of thousands of training scenarios, and billions of training iterations,
> 3) proposing novel utility functions, evaluating new and existing ones across multiple scaling dimensions, and analyzing their correlation with each other and performance metrics.
>
> We agree that achieving the contributions listed above requires significant engineering effort; hence, we understand why our paper may appear to be only a benchmarking/application paper. However, we think these are valuable contributions, as automated curricula have been studied only a few times in the autonomous driving context and have not been attempted at scale. Therefore, our work paves the way for future research on curriculum learning algorithms in industry-scale simulators.
>
> Regarding whether the proposed utility functions outperform existing ones, we do not make such a claim in our paper. We showcase that
> 1) Realism-based functions, which are commonly used in evaluating AD policies, can effectively prioritize scenarios to accelerate learning,
> 2) Learnability-hard, which relies on a more difficult-to-satisfy success criterion compared to learnability, can match the performance of learnability in general, despite depending on a sparser success signal.
>
> The recently added discussion on the correlation between utility functions and performance metrics clarifies why some functions are more performant than the rest, e.g., the comparison between Act-MAE and GC-ADE (see Section 5.5).
>
> We would also like to clarify that we introduce the background (Section 3) and the method (Section 4) separately. The background section is quite detailed, mainly because CL4AD integrates many concepts into a batched AD simulator: underspecified partially observable stochastic games, self-play RL, unsupervised environment design, and, as an instance of it, prioritized level replay. We wanted to set the stage for all these concepts before introducing CL4AD. The reason is that CL4AD adapts UED methods and utility functions, which are traditionally designed for single-agent and single-world RL, to the batched self-play setting.
>
> **W2**: We agree with the reviewer that GIGAFLOW's zero-shot generalization to widely-used benchmarks shows a real-world impact that a batched AD simulator can achieve. Nevertheless, GIGAFLOW (closed-source) and GPUDrive (open-source) are quite different simulators. GIGAFLOW trains on 8 GPUs, with 4800 worlds per GPU, which is 48 times more than what GPUDrive enables, as it currently runs on a single GPU, with 800 worlds. In addition, GIGAFLOW has a quite complex reward function with 10 components that address comfort, lane keeping, speed limits, and more. In contrast, GPUDrive has only three components that cover goal completion and collision/off-road avoidance. Finally, GIGAFLOW achieves this real-world impact after 10 days of training with significantly more compute power, whereas we train our agents for about 2.5 days.
>
> Our work showcases the out-of-distribution generalization capabilities of curriculum learning at varying scales by evaluating trained agents in unseen traffic scenarios using real-world datasets used to benchmark AD models. Furthermore, we show another instance of real-world impact: curriculum learning reduces wall-clock time to achieve a 99% success rate by up to 1 billion interactions, which amounts to more than a day of training.
>
> **W3**: We thank the reviewer for suggesting these papers. We have included [1] to exemplify the use of PLR for meta-RL, where the number of training tasks scales up to 1000 (see discussion on curriculum learning for RL in Section 2), and [2] to evidence that heuristic curriculum has been shown to have benefits in an AD simulator (see discussion on curriculum learning for AD in Section 2).
>
> **Reference**:
> 1) Matthew T Jackson, Minqi Jiang, Jack Parker-Holder, Risto Vuorio, Chris Lu, Greg Farquhar, Shimon Whiteson, and Jakob Foerster. Discovering general reinforcement learning algorithms with adversarial environment design. Advances in Neural Information Processing Systems, 36:79980– 79998, 2023.
> 2) Quanyi Li, Zhenghao Mark Peng, Lan Feng, Zhizheng Liu, Chenda Duan, Wenjie Mo, and Bolei Zhou. Scenarionet: Open-source platform for large-scale traffic scenario simulation and modeling. Advances in neural information processing systems, 36:3894–3920, 2023.

---

### Official Review · Reviewer_hTh4 · 2025-11-01

**Soundness:** 3
**Presentation:** 3
**Contribution:** 2
**Rating:** 6
**Confidence:** 3

**Summary:**

This paper addresses the sample inefficiency of training autonomous driving (AD) agents in modern, large-scale batched simulators. While simulators like GPUDRIVE achieve massive throughput (billions of interactions per day), the standard training method, domain randomization (DR) or uniform sampling, wastes compute on scenarios that are either too easy or too hard.

The authors propose CL4AD, the first framework to integrate and scale curriculum learning (CL) within such a batched, self-play simulator. The core idea is to frame scenario selection as an Unsupervised Environment Design (UED) problem. The system uses Prioritized Level Replay (PLR) as its backbone, which adaptively samples scenarios from a large dataset (Waymo Open Motion Dataset) based on their "utility" for learning.

**Strengths:**

- The paper tackles a clear and significant bottleneck. As simulation throughput increases, sample efficiency (i.e., what to simulate) becomes the next logical and practical problem to solve. This work is at the forefront of this shift.
- The performance gains are not minor; they are massive. Achieving a 99% success rate 1 billion steps earlier (or 77% faster in wall time) is a very strong and convincing result. The paper does an excellent job of validating this at different scales (1k, 10k, 80k scenarios) and under compute constraints, showing the method is robust.
- While the core CL algorithm (PLR) is an existing method, its application and integration into a batched, multi-agent, self-play simulator at this scale is a non-trivial and valuable engineering contribution. This is the first work to successfully do so.

**Weaknesses:**

- The primary weakness is that the paper does not propose a new curriculum learning algorithm. The core method is PLR, which is from 2021. The contribution is one of integration, scaling, and new utility functions, making this more of a systems and application paper than a fundamental RL methods paper.
- The paper is titled "Scaling Curriculum Learning," but it only scales one specific UED algorithm (PLR). It does not explore other prominent UED methods cited in its own related work, such as ACCEL or RE-PAIRED, which might have different scaling properties or performance characteristics.
- The only baseline used for comparison is Domain Randomization (DR). While DR is the current standard and thus the most important baseline, the paper would be stronger if it compared against other, simpler curriculum heuristics (e.g., replaying failed episodes, or an annealed curriculum based on scenario complexity like the number of agents) to better situate the gains from a sophisticated method like PLR.

**Questions:**

- The algorithmic contribution is primarily the integration of PLR. Why was PLR chosen over other UED methods like ACCEL, which also scales and mutates scenarios to create new ones (as mentioned in your limitations)?
- Your realism-based utility functions (UGC-ADE, UAct-MAE) are a key contribution. Did you observe any qualitative differences in the policies trained with these metrics versus regret-based metrics? For example, did they result in more "human-like" (but perhaps less optimal) driving behavior?

---

> ### Author Response · Authors · 2025-11-27
>
> We would like to thank the reviewer for their thought-provoking comments and questions that helped us improve our work in the rebuttal phase. Below, we first respond to their comments regarding the weaknesses, and then to their questions.
>
> **W1**: The reviewer is correct that the contribution of CL4AD is in the form of
> 1) integration of curriculum learning into a batched AD simulator for the first time,
> 2) scaling CL for tens of controllable agents in a single scenario, hundreds of concurrent worlds, tens of thousands of training scenarios, and billions of training iterations,
> 3) proposing novel utility functions, evaluating new and existing ones across multiple scaling dimensions, and analyzing their correlation with each other and performance metrics.
>
> We agree that achieving the contributions listed above requires significant engineering effort; hence, we understand why our paper may appear to be only a systems/application paper. However, we think these are valuable contributions, as automated curricula have been studied only a few times in the autonomous driving context and have not been attempted at scale. Therefore, our work paves the way for future research on curriculum learning algorithms in industry-scale simulators.
>
> **W2**: Indeed, we focus on scaling PLR because it is the backbone of many UED methods and has been shown to scale (in different ways) outside the autonomous driving context (see the paragraph on curriculum learning for RL in Section 2). However, in practice, we study more than one algorithm: Regret-based utility functions, positive value loss, and maximum Monte Carlo come from Robust PLR [1], and success-based utility function learnability comes from Sampling for Learnability (SFL) [2]. Robust PLR and SFL have almost identical buffer and sampling mechanisms with PLR, except that Robust PLR does not update the policy using rollouts from unseen levels, and SFL has a filtering mechanism that requires additional rollouts to assess whether a level has high learnability. Although we decided to omit such a discussion, our work essentially investigates how Robust PLR, SFL, and PLR scale. We thank the reviewer for raising this point. We have added a brief discussion to Sections 3.4 and 5 to clarify this aspect.
>
> As the discussion on future work in Section 6 briefly notes, we have not studied ACCEL and RE-PAIRED because these approaches do not merely curate existing levels but also edit them or generate entirely new ones. Mutating scenarios involves removing/adding controllable agents, changing the initial/goal positions of existing agents, changing vehicle attributes, etc. [3,4]. Although [3,4] adapt RE-PAIRED and ACCEL to the AD setting, they do so in the niche context of urban interactions, with a handful of road layouts and a few vehicles per scenario. Scaling editing/mutation algorithms while ensuring that synthetically generated scenarios are realistic and feasible is not a trivial task and takes the focus away from curriculum learning. In fact, in the context of autonomous driving, synthetic scenario generation is a research topic on its own [5-9]. Recently, [10] proposed ScenarioDreamer, which synthetically generates initial scenes with state-of-the-art speed and performance. Therefore, in our future work, we plan to focus on curriculum-guided synthetic scenario generation for batched AD simulators, enabling comparisons with baselines such as random mutation and state-of-the-art generative models like ScenarioDreamer.
>
> **W3**: We thank the reviewer for suggesting this idea, which helps showcase why automated curriculum methods such as PLR are preferable to heuristic-based counterparts. We have followed their advice and evaluated two heuristic-based curriculum methods that prioritize scenarios based on a static utility: the number of controllable agents in a scenario. We refer the reviewer to the global response for a summary of these baselines and section 5.1 for the newly added results in the updated manuscript.

---

> ### Author Response · Authors · 2025-11-27
>
> **Q1**: Please see our response to **W2**, where we cover this issue and more.
>
> **Q2**: Regardless of the utility functions, all agents are trained with the same reward function, where the agent is incentivized to reach the goal as quickly as possible while avoiding collisions and going off-road. As a result, all policies are trained to optimize the same objectives. The utility functions act as proxies to identify scenarios that can accelerate the agent's learning, not what it learns. Even if the utility functions can prioritize different scenarios, the agent will still learn to maximize the reward function. Therefore, in general, there are no significant differences between scenarios in terms of realism. Figure 4, which visualizes the progression of GC-ADE (i.e., the distance to logged trajectories), shows that different utility functions converge to similar values.
>
> **References:**
> 1) Minqi Jiang, Michael Dennis, Jack Parker-Holder, Jakob Foerster, Edward Grefenstette, and Tim Rocktaschel. Replay-guided adversarial environment design. Advances in Neural Information Processing Systems, 34:1884–1897, 2021a.
> 2) Alexander Rutherford, Michael Beukman, Timon Willi, Bruno Lacerda, Nick Hawes, and Jakob Foerster. No regrets: Investigating and improving regret approximations for curriculum discovery. Advances in Neural Information Processing Systems, 37:16071–16101, 2024.
> 3) Brunnbauer, A., Berducci, L., Priller, P., Nickovic, D., & Grosu, R. (2024). Scenario-based curriculum generation for multi-agent autonomous driving. arXiv preprint arXiv:2403.17805.
> 4) Abouelazm, A., Weinstein, T., Joseph, T., Schörner, P., & Zöllner, J. M. (2025). Automatic Curriculum Learning for Driving Scenarios: Towards Robust and Efficient Reinforcement Learning. arXiv preprint arXiv:2505.08264.
> 5) Lu, J., Wong, K., Zhang, C., Suo, S., & Urtasun, R. (2024, May). Scenecontrol: Diffusion for controllable traffic scene generation. In 2024 IEEE International Conference on Robotics and Automation (ICRA) (pp. 16908-16914). IEEE.
> 6) Tan, S., Wong, K., Wang, S., Manivasagam, S., Ren, M., & Urtasun, R. (2021). Scenegen: Learning to generate realistic traffic scenes. In Proceedings of the IEEE/CVF Conference on Computer Vision and Pattern Recognition (pp. 892-901).
> 7) Mi, L., Zhao, H., Nash, C., Jin, X., Gao, J., Sun, C., ... & Anguelov, D. (2021). Hdmapgen: A hierarchical graph generative model of high definition maps. In Proceedings of the IEEE/CVF Conference on Computer Vision and Pattern Recognition (pp. 4227-4236).
> 8) Chitta, K., Dauner, D., & Geiger, A. (2024, September). Sledge: Synthesizing driving environments with generative models and rule-based traffic. In European Conference on Computer Vision (pp. 57-74). Cham: Springer Nature Switzerland.
> 9) Sun, S., Gu, Z., Sun, T., Sun, J., Yuan, C., Han, Y., ... & Ang Jr, M. H. (2023). Drivescenegen: Generating diverse and realistic driving scenarios from scratch. arXiv preprint arXiv:2309.14685.
> 10) Rowe, L., Girgis, R., Gosselin, A., Paull, L., Pal, C., & Heide, F. (2025). Scenario dreamer: Vectorized latent diffusion for generating driving simulation environments. In Proceedings of the Computer Vision and Pattern Recognition Conference (pp. 17207-17218).

---

### Official Review · Reviewer_496G · 2025-11-01

**Soundness:** 3
**Presentation:** 3
**Contribution:** 2
**Rating:** 4
**Confidence:** 3

**Summary:**

- This paper proposes Curriculum Learning for Autonomous Driving (CL4AD) for batch AD simulators.
- The authors frame the CL problem as an unsupervised environment design (UED) problem. They try to improve upon random sampling of the levels/scenarios/environment for the batched sims for RL training.
- The authors introduce utility functions to score the scenarios/level of curricula to shape the replay distribution.
  - The scores are dependent on regret, success and realism of the AD  trajectories.
  - They combined the prioritized level replay (PLR) to adaptively sample high utility scenarios with different scoring functions and analyse their effects on the overall AD policy performance.
- With the introduction of curriculum, the large scale experiments claimed to achieve 99% success in GPUDrive a billion steps earlier than random sampling and also reduced the wall clock time by 77%.
- The authors perform experiments to find if CL can accelerate learning in AD policies, is it effective under limited compute resources and can it be scaled with more scenarios.

It is appreciated that while the authors have proposed a good integration of CL for batched simulators and presented multiple experiments and ablations to support the approach, it would be more advantageous if there were some explicit analysis done on why a particular utility function is better or worse than the other and which one can be preferred under which conditions.

Given the comments related to weaknesses and limitations, I can have the rating as 4. Flexible to move this during or after rebuttal.

**Strengths:**

- The paper did a good job laying out the theoretical background required for integrating UED based CL for AD.
- The paper presents three novel utility functions from which UAct-MAE (a realism metric calculating distance between RL agents’ actions and logged trajectories) had relatively better results in terms of returns and success rate.
- The paper visualized the evolution of the utility score values (replay distributions) with the number of policy updates for different scoring functions. This helps in analysing how those values evolve as the policy improves with increasing steps.
- The paper is able to bring the advantages of CL in terms of sample efficiency for training AD policies using batched simulators.
- The approach seems to be scalable and extensible and can be applied to other simulators as well.
- A well written paper with clear figures, plots and diagrams and ablations.

**Weaknesses:**

1. The plots were not clear enough in showing how the utility score evolved with training. They are of almost the same color throughout the number of policy updates.

2. No analysis or comments on why a certain utility score function is performing relatively better than the other.


**Limitations**
- There can be other study/experiments related to heuristic-based curriculum design as in increasing the number of agents, different traffic densities, etc as a proxy for difficulty levels. This can be compared to the automated curriculum.
- The paper might be strengthened by framing this as a designs-space formulation on how CL for AD and directly used a plug-and-play option for batched simulators.

**Questions:**

1. Line 151: Definition 3.2 - The POSG is underspecified. Can we have some examples of what could be some unknown components or uncertainties that are making the observations underspecified?

2. Fig 5: We might assume more scenarios with high utility scores in the beginning of the training and a gradual decrease with more updates as the policy improves. However, such a trend was not observed from the plot (first row) and the scores seem to be pretty consistent as the number of policy update steps increased.

3. Fig 5 (middle row): Would that mean scenarios that have around > 10-15 agents are not useful for learning.
  - Either they are too easy to effect the utility score, or
  - Too difficult to match the score to get any useful signal from it.

4. How are the scenarios/levels evicted from the buffer?

5. What is the general update-to-data ratio (UTD ratio) for policy training?

**Suggestion:**
- It would be helpful to have some of the acronyms used in the figures to have their full-form in the figure description. It could be confusing as some figures appear way before the text that introduces those terms (for example Fig 2).
- It would be nice to have configuration of the replay buffer (sample to insert ratio, eviction policy, etc.)

---

> ### Author Response · Authors · 2025-11-27
>
> We would like to thank the reviewer for their positive comments, valuable criticism, and thought-provoking questions that helped us improve the presentation of our work. We first respond to the reviewer's comments, then their questions, and finally their suggestions.
>
>
> **W1**: To clarify the message that plots of utility progression convey, we have replaced them with box plots with fewer policy updates on the x-axis (see Figures 4 and 6 in the updated manuscript).
>
> **W2**: To address the reviewer's request for a discussion of utility functions and their performance, we have provided a comprehensive analysis of their correlation with each other and performance metrics (see Section 5.5 of the updated manuscript and our global response for a summary).
>
> **L1**: We thank the reviewer for suggesting this idea. We have experimented with heuristic-based curricula based on the number of agents in a scenario. Please check Section 5.1 for the corresponding results and our global response for its summary.
>
> **L2**: Could the reviewer please clarify what they mean by design-space? Our understanding is that it covers the choice of utility functions and even mechanisms for mutating or generating scenarios as part of curriculum learning. As the paragraph on future work in Section 6 indicates, we are planning to extend our work to investigate methods to mutate existing scenarios or synthetically generate new ones based on prioritized attributes, namely, road layouts, traffic density, behavior types, etc.. Such a study will focus on identifying critical properties of scenarios, their relation to utility functions, and how to increase scenario diversity to improve robustness. In fact, synthetic traffic scenario generation is a research topic on its own [1-10].
>
> **Q1**: In practice, level $\theta\in\Theta=[M]$ is a unique ID, where $M$ is the number of scenarios in the dataset; hence, it does not represent the attributes of the corresponding scenario. For example, the road layout, whether the scenario is signalized, and the number of vehicles are not specified, hence neither the level generator nor the student has access to this information based on $\theta$ alone. In addition, the agent does not observe $\theta$ at all; thus, the agent does not know which scenario it is in. An agent only observes its surroundings, i.e., a partial view based on sensor readings. We have clarified Section 3.1 to address this question.
>
> **Q2**: We would like to clarify that the color corresponds to the probability of a scenario ID (first row) or the number of vehicles (middle row) based on the replay distribution, which is a combination of score ranking distribution, see Eq. (6), and staleness distribution, see Eq. (7). Therefore, evolution of probabilities do not correspond to evolution of scores. As the contribution of the staleness distribution is generally low, due to $\rho\in\{0.1,0.3\}$ in Eq. (5), we may interpret Fig. 5 as the evolution of score ranking. Therefore, we observe that specific scenario IDs consistently rank high with respect to their utility, which is $U^{\text{MaxMC}}$ in Fig. 5, not that their utility scores stay high. We have made a small change in Figure 5 to indicate where the prioritized scenarios fall on the y-axis in both the top and middle rows.
>
>
> **Q3**: The middle row in Fig. 5 is a distribution where the replay probabilities are grouped with respect to the number of vehicles in scenarios. This distribution is heavily biased by the dataset, as most scenarios involve fewer than 10 controllable agents. However, when combined with $U^\text{MaxMC}$, PLR still prioritizes specific scenarios with more than 15 agents, e.g., scenarios 58 and 264 (bottom row). Therefore, even in crowded scenarios, learning can still be useful.
>
> An interesting aspect of PLR is that the score temperature $\beta$ (see Eq. (6)) determines the uniformity of the score ranking distribution. A more uniform score-ranking distribution, in terms of probabilities of scenario IDs, is more biased towards the dataset. Given low $\beta=2$, e.g., $U^\text{MaxMC}$, $U^\text{Act-MAE}$, and $U^\text{Learn-Hard}$, see Figs. 5, 12, and 15, respectively, certain scenarios dominate the replay distribution (see a handful of very dark red IDs in each column of the first row). As a result, the distribution in the middle row evolves during training, even though it may not be as drastic as the first row. Given a high $\beta=4$, the distribution in the middle row largely stays similar, see Figs. 14, 15, 17, and 18, as they are more biased towards the dataset due to their lesser impact on score rankings.

---

> ### Author Response · Authors · 2025-11-27
>
> **Q4**: In our experiments, the buffer length is set to the dataset size, so no scenarios are evicted. For cases where the buffer is smaller than the dataset, if a new scenario has higher utility than the lowest-utility scenario in the buffer, the new scenario replaces the existing one. We have added a clarifying sentence to Appendix D that explains how PLR operates in a batched AD simulator.
>
> **Q5**: Table 1 in Appendix E.2. describes parameters related to policy training. In cases 1-3, every 524,288 interactions, the self-play policy is updated using the experience buffer, which contains only interactions since the last policy update. Hence, during a training for 2 billion steps, the policy gets updated over 3800 times. In the ablation study on compute constraints, the experience buffer size drops down to 131,072, where the total number of policy updates is around 7600.
>
> We sample scenarios from the curriculum every 2 million steps. For cases 1-3, this corresponds to a curriculum sample every 3-4 policy updates, whereas for the ablation study, it is around 15 policy updates.
>
> **Suggestion 1**: To clarify the acronyms in Figure 2, we have updated its caption to direct the reader to the relevant sections. Although we agree that a complete description of acronyms would make it clearer, it would also take more than a few lines. We placed Figure 2 in the introduction to provide evidence that PLR outperforms DR early in the paper. Our intention was not to discuss how utility functions differ in the introduction.
>
> **Suggestion 2**: We have added more explanation in Section 4 and Appendix D to provide more details on sampling from and updating the PLR buffer.
>
>
> **References:**
> 1) Lu, J., Wong, K., Zhang, C., Suo, S., & Urtasun, R. (2024, May). Scenecontrol: Diffusion for controllable traffic scene generation. In 2024 IEEE International Conference on Robotics and Automation (ICRA) (pp. 16908-16914). IEEE.
> 2) Tan, S., Wong, K., Wang, S., Manivasagam, S., Ren, M., & Urtasun, R. (2021). Scenegen: Learning to generate realistic traffic scenes. In Proceedings of the IEEE/CVF Conference on Computer Vision and Pattern Recognition (pp. 892-901).
> 3) Mi, L., Zhao, H., Nash, C., Jin, X., Gao, J., Sun, C., ... & Anguelov, D. (2021). Hdmapgen: A hierarchical graph generative model of high definition maps. In Proceedings of the IEEE/CVF Conference on Computer Vision and Pattern Recognition (pp. 4227-4236).
> 4) Chitta, K., Dauner, D., & Geiger, A. (2024, September). Sledge: Synthesizing driving environments with generative models and rule-based traffic. In European Conference on Computer Vision (pp. 57-74). Cham: Springer Nature Switzerland.
> 5) Sun, S., Gu, Z., Sun, T., Sun, J., Yuan, C., Han, Y., ... & Ang Jr, M. H. (2023). Drivescenegen: Generating diverse and realistic driving scenarios from scratch. arXiv preprint arXiv:2309.14685.
> 6) Rowe, L., Girgis, R., Gosselin, A., Paull, L., Pal, C., & Heide, F. (2025). Scenario dreamer: Vectorized latent diffusion for generating driving simulation environments. In Proceedings of the Computer Vision and Pattern Recognition Conference (pp. 17207-17218).

---

### Author Response · Authors · 2025-11-27
**Global Response: New baselines, discussions, results, visualizations**

We appreciate all reviewers for their thought-provoking comments and helpful questions. Thanks to them, we added new baselines, discussions, results, and visualizations to support our claims better. Below, we list the changes we made:

- Added two heuristic-based curriculum methods, Heuristic-Dense and Heuristic-Sparse, that prioritize traffic scenarios with high and low traffic density, respectively (see Section 5.1),
- Extended the results in case 2 with the rest of the utility functions (see Section 5.4),
- Added a third seed to case 3, where we train with the largest training dataset (80,000 scenarios) (see Section 5.4),
- Provided a comprehensive analysis on the correlation between utility functions and performance metrics (see Section 5.5),
- Modified Figure 1, which visualizes the workflow of CL4AD, so that it illustrates each category of utility functions (see Section 1)
- Updated the related work with suggested papers on curriculum learning (see Section 2)
- Made clarifications on underspecified POSGs and how PLR relates to existing UED methods (see Sections 3.1, 3.4, and 5)
- Added details on how PLR updates its level buffer (see Section 4 and Appendix D)
- Replaced line plots for progression of utility with box plots (see Figures 4 and 6)

We would like to ask the reviewers to read the updated manuscript (changes are highlighted in red). Below, we provide summaries for the two main extensions we have made:

1) Evaluation of heuristic-based curricula

    At every curriculum sample call, with probability p=0.5, a heuristic-based curriculum uniformly samples scenarios from the dataset; otherwise, it samples from a score-rank distribution in which the utility of a scenario is a function of its traffic density. We have enabled two options, such that either dense (Heuristic-Dense) or sparse (Heuristic-Sparse) scenarios rank high. Note that, as the utility of a scenario is based on its inherent properties, i.e., how many agents exist in it, the utility score is fixed. In contrast, PLR methods have adaptive scores for each scenario, as the utility is a function of the trained policy. Our experiments in case 1 showed that, compared with Heuristic-Sparse and Heuristic-Dense, PLR reduces wall-clock time to achieve a 99% success rate by 40% and 66%, respectively. We extended our experiments to case 3, where Heuristic-Dense can match the sample-efficiency of PLR, yet Heuristic-Sparse does not bring any advantages over DR. These results showcase that even though heuristic-based curricula can accelerate training, as it is not adaptive, its performance is not as consistent as PLR, which is an automated curriculum method.

2) Comprehensive analysis of the correlation between utility functions and performance metrics

    Section 5.5 provides a heat map of Pearson correlations within and across categories of utility functions, as well as with performance metrics. We investigate each case separately to determine whether scaling up the dataset affects results and evaluate policies from all checkpoints of the training to include agents with varying capabilities. A detailed discussion is available in the corresponding section.

    Utility functions correlate strongly within their own categories in general: regret-based utilities are highly positively correlated (and increasingly so with more training data), and success-based utilities also align closely. In contrast, realism-based utilities do not correlate with one another.

    Across categories, regret and success utilities show a mild positive correlation because high-regret scenarios often exhibit high success variance. Realism remains mostly uncorrelated, as, in short, optimality with respect to the reward function in GPUDrive does not correspond to realistic behavior.

    Correlations with performance metrics vary: AMGAE correlates with collision/off-road rates (explaining its poor performance in case 1), PVL correlates with none, MaxMC correlates positively with returns (but weakens across cases), and success-based utilities correlate negatively with return and success. Realism-based utilities show no consistent correlation with performance.

---

### Meta-Review · Area_Chair_8zzZ · 2026-01-05

**Summary:**

This paper studies curriculum learning for batched autonomous driving simulators (GPUDrive) by framing scenario selection as UED and scaling PLR-style replay with several utility functions (regret/success/realism). Reviewers agree the work is sound, clearly written, and shows gains in sample efficiency and wall-clock time. The rebuttal improves clarity (better figures, added implementation details), adds heuristic curriculum baselines, and provides additional analysis of utility-function correlations and overhead. However, the primary concern remains that the core approach is largely an integration and benchmarking of existing PLR-based methods rather than a new algorithmic contribution, and the scope of evaluation remains centered on a single simulator/dataset setup without broader validation on widely used AD benchmarks.

AC has checked the submission, the reviews, and the rebuttal, and AC considers this a borderline paper and AC has spent more time evaluating it. Unfortunately, given the moderate novelty and positioning closer to a systems/application study than a new methodological advance, the AC recommends rejection.

**Reviewer Concerns:**

Concerns addressed by the rebuttal:
(1) Lack of analysis on utility functions and when/why they work (496G, rqJS)
Authors added Section 5.5 analyzing correlations between utility functions and performance metrics, clarifying redundancy/correlation and providing interpretation for differences (e.g., Act-MAE vs GC-ADE).

(2) Missing heuristic curriculum baselines (496G, hTh4)
Authors added heuristic curricula based on number of agents (static difficulty proxy) and incorporated results in Section 5.1, positioning PLR-based curricula more clearly.

(3) Confusing/unclear plots of utility evolution (496G)
Authors replaced plots with box plots and improved figure clarity (Figures 4 and 6) and clarified interpretation that the figure shows replay probabilities/rankings rather than raw score evolution.

Still outstanding / partially addressed:
(1) Core novelty remains moderate (hTh4, oqQS, rqJS, 496G)
Even after rebuttal, the central critique remains: the paper does not propose a new CL/UED algorithm, and the new utility functions are useful but not consistently outperforming prior PLR utilities.

(2) Evaluation on additional public benchmarks (oqQS, rqJS)
Reviewers specifically ask for evaluation on broader AD benchmarks (e.g., WOSAC/nuPlan). Authors respond that WOMD is a widely used public benchmark and emphasize the submission area is application-focused.

(3) “Scaling curriculum learning” framing scope (hTh4)
The title/positioning suggests scaling CL broadly, but the study primarily scales PLR-style approaches. Authors clarify scope and mention related variants, but it remains a narrower interpretation of “scaling CL.” Good to narrow the claim of this work.

**Reviewer Scores:**

* Reviewer 496G (initial 4): Likely score change 4 → a slightly higher score.
* Reviewer hTh4 (initial 6, positive on impact; novelty critique). Key issues: no new algorithm, narrow baselines, why PLR over other UED. Likely remains as 6
* Reviewer oqQS (initial 4, strongest novelty skepticism + asks broader benchmarks + missing related work). Likely score change: 4 → 4, and championing the rejection.
* Reviewer rqJS (initial 6, and explicitly maintained score after rebuttal). They acknowledged rebuttal improvements but reiterated that it is more application than fundamental RL and cited the lack of WOSAC-style benchmarks; they kept their score.

---

### Decision · Program_Chairs · 2026-01-26

Reject